# Unimodal Likelihood Models for Ordinal Data

**Ryoya Yamasaki**                                                          *yamasaki@sys.i.kyoto-u.ac.jp*
*Department of Systems Science*
*Graduate School of Informatics, Kyoto University*
*36-1 Yoshida-Honmachi, Sakyo-ku, Kyoto 606-8501 JAPAN*

**Reviewed on OpenReview:** *https://openreview.net/forum?id=1lOsClLiPc*

## Abstract

Ordinal regression (OR) is the classification of ordinal data, in which the underlying target variable is categorical and considered to have a natural ordinal relation for the explanatory variables. In this study, we suppose the unimodality of the conditional probability distribution of the target variable given a value of the explanatory variables as a natural ordinal relation of the ordinal data. Under this supposition, unimodal likelihood models are considered to be promising for achieving good generalization performance in OR tasks. Demonstrating that previous unimodal likelihood models have a weak representation ability, we thus develop more representable unimodal likelihood models, including the most representable one. OR experiments in this study showed that the developed more representable unimodal likelihood models could yield better generalization performance for real-world ordinal data compared with previous unimodal likelihood models and popular statistical OR models having no unimodality guarantee.

## 1 Introduction

Ordinal regression (OR, also called ordinal classification) is the classification of ordinal data, in which the underlying target variable is categorical and considered to have a natural ordinal relation for the explanatory variables; see Section 2 for a detailed formulation. Typical examples of the target label set of ordinal data are sets of grouped continuous variables like age groups {'0 to 9 years old', '10 to 19 years old', ..., '90 to 99 years old', 'over 100 years old'} and sets of assessed ordered categorical variables like human rating {'strongly agree', 'agree', 'neutral', 'disagree', 'strongly disagree'} (Anderson, 1984), and various practical tasks have been tackled within the OR framework: for example, face-age estimation (Niu et al., 2016; Cao et al., 2019; Anonymous, 2022), information retrieval (Liu, 2011), credit or movie rating (Kim & Ahn, 2012; Yu et al., 2006), and questionnaire survey in social research (Chen et al., 1995; Bürkner & Vuorre, 2019).

Consider an example of the questionnaire survey about support for a certain idea that requires subjects to respond from {'strongly agree', 'agree', ...}. Here, it would seem possible that subjects, who have features specific to those who typically respond 'agree', respond 'neutral', but unlikely that they respond 'disagree'. Such a phenomenon in OR tasks can be rephrased as the unimodality of the conditional probability distribution (CPD) of the underlying target variable given a value of the underlying explanatory variables. The hypothesis "many statisticians or practitioners often judge that the data have a natural ordinal relation and decide to treat them within the OR framework, with unconsciously expecting their unimodality" may be convincing, as we will experimentally confirm in Section 6.2 that many ordinal data, treated in previous OR studies that do not consider the unimodality, tend to have a unimodal CPD.

Commonly, the generalization performance of a statistical model or classifier (OR method) based on that model depends on the underlying data distribution and the representation ability of that model. Recall that the generalization performance can be roughly decomposed into bias- and variance-dependent terms (well-known bias-variance decomposition); a model in which the representation ability is too strongly restricted will result in a large bias-dependent term if it cannot represent the underlying data distribution, and a model

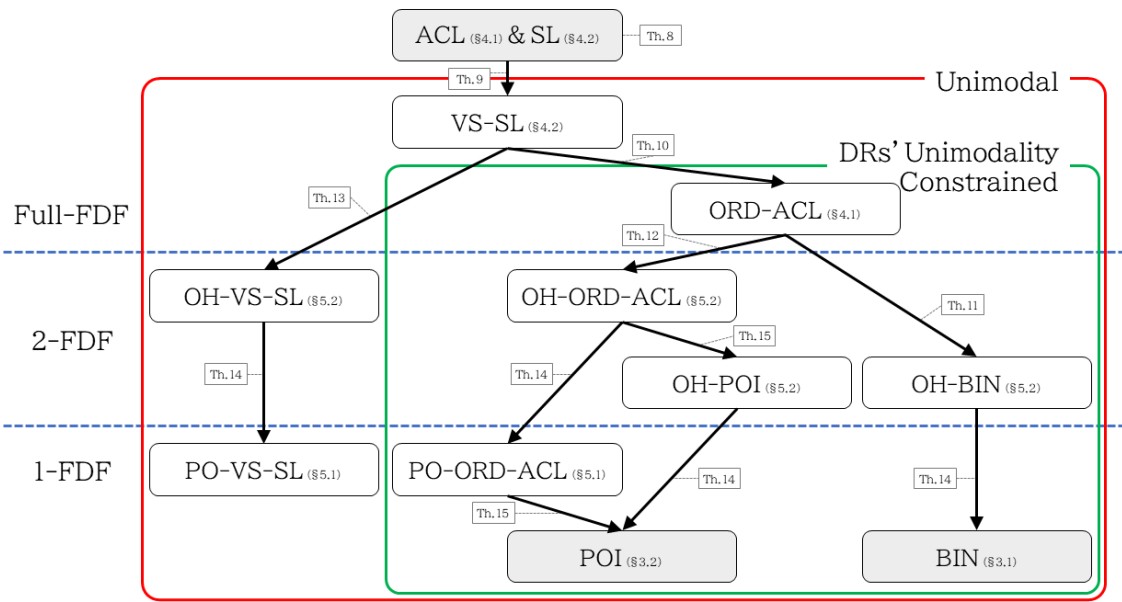

Figure 1: Relationship among the representation abilities of 12 likelihood models that we consider in this paper (except for Section 5.3). We describe the name of a model and the section number of the section in which the model is studied in a rounded box. Models in rounded boxes with gray background are previous ones, and models in rounded boxes with white background are proposed in this paper. The directed graph with 'rounded box'-shaped nodes implies that models in a parent node are more representable than models in a child node in the sense of Definition 5. Also, we describe the theorem number of the theorem that shows the relationship in a square box.

that is unnecessarily flexible to represent the underlying data distribution will result in a large variance-dependent term especially for a small-size training sample, and such models at both extremes may yield bad generalization performance. Therefore, assuming the unimodality of the CPD as a natural ordinal relation of ordinal data, unimodal likelihood models, which can adequately represent such unimodal data and are compact with respect to the representation ability than unconstrained statistical models such as multinomial logistic regression model, are considered to be promising for achieving good generalization performance in the conditional probability estimation and OR tasks.

The existing studies (da Costa et al., 2008; Iannario & Piccolo, 2011; Beckham & Pal, 2017) were inspired by the shape of the probability mass function (PMF) of elementary categorical probability distributions such as binomial, Poisson, and uniform distributions and developed unimodal likelihood models. In this paper, we introduce several notions that characterize the shape of CPDs in Section 2.1 and notions that characterize the representation ability of likelihood models in Section 2.2. Then, on the basis of these notions, we clarify characteristics of data distributions that their likelihood models can not represent well and show that their likelihood models have a weak representation ability, in Section 3. We thus propose more representable unimodal likelihood models in Sections 4 and 5. Those proposed unimodal likelihood models bridge the gap between previous unconstrained likelihood models (e.g., ACL and SL models in Figure 1) that have no unimodality guarantee and may be unnecessarily flexible and previous unimodal likelihood models (e.g., POI and BIN models in Figure 1) that may be too weakly representable. In particular, VS-SL model described in Section 4.2 is the most representable one among the class of unimodal likelihood models.

We performed experimental comparisons of 2 previous unimodal likelihood models, 2 popular statistical OR models without the unimodality guarantee, and 8 proposed unimodal likelihood models; see Section 6 and Appendix C. Our empirical results show that the proposed more representable unimodal likelihood models can be effective in improving the generalization performances for the conditional probability estimation and OR tasks for many data that have been treated in previous OR studies as ordinal data. On the basis of these

results, this study suggests the effectiveness of the proposed unimodal likelihood models and OR methods based on those models.

## 2 Preliminaries

### 2.1 Ordinal Regression Tasks and Ordinal Data

The OR task is a classification task. Denoting explanatory and categorical target variables underlying the data as $\boldsymbol{X} \in \mathbb{R}^d$ and $Y \in [K] \coloneqq \{1, \ldots, K\}$, we formulate the OR task as searching for a classifier $f : \mathbb{R}^d \to [K]$ that is good in the sense that the task risk $\mathbb{E}[\ell(f(\boldsymbol{X}), Y)]$ becomes small for a specified task loss $\ell : [K]^2 \to [0, \infty)$, where the expectation $\mathbb{E}[\cdot]$ is taken for all included random variables (here $\boldsymbol{X}$ and $Y$). Popular task losses in OR tasks include not only the zero-one loss $\ell_{\mathrm{zo}}(j, k) \coloneqq \mathbb{1}\{j \neq k\}$, where $\mathbb{1}\{c\}$ values 1 if a condition $c$ is true and 0 otherwise, but also V-shaped losses reflecting one's preference of smaller prediction errors over larger ones such as the absolute loss $\ell_{\mathrm{abs}}(j, k) \coloneqq |j - k|$ and the squared loss $\ell_{\mathrm{sq}}(j, k) \coloneqq (j - k)^2$.

In the OR framework, it is supposed that the underlying categorical target variable $Y$ of the data is equipped with an ordinal relation naturally interpretable in the relationship with the underlying explanatory variables $\boldsymbol{X}$, like examples described in the head of Section 1. We here assume that the target labels are encoded to $1, \ldots, K$ in an order-preserving manner, like from 'strongly agree', 'agree', 'neutral', 'disagree', 'strongly disagree' to $1, \ldots, 5$ or $5, \ldots, 1$. The OR framework considers the classification of such ordinal data. Note that, like most previous OR studies have discussed the OR without formal common understanding of what constitutes ordinal data and their natural ordinal relation, it is difficult to define ordinal data any more rigorously, and we in this paper refer to the data discussed in previous OR studies as ordinal data.

As we declared in Section 1, this study basically assumes, as a natural ordinal relation of ordinal data, the unimodality in the theoretical discussion or the almost-unimodality to real-world ordinal data, precisely defined in the following:

**Definition 1.** *For a vector $\boldsymbol{p} = (p_k)_{k \in [K]} \in \mathbb{R}^K$, we define $M(\boldsymbol{p}) \coloneqq \min(\arg\max_k (p_k)_{k \in [K]})^1$, and say that $\boldsymbol{p}$ is unimodal if it satisfies*

$$p_1 \leq \cdots \leq p_{M(\boldsymbol{p})} \ and \ p_{M(\boldsymbol{p})} \geq \cdots \geq p_K. \tag{1}$$

*Also, we call $M(\boldsymbol{p})$ the mode of $\boldsymbol{p}$ if $\boldsymbol{p}$ is a PMF satisfying $\boldsymbol{p} \in \Delta_{K-1}$, where $\Delta_{K-1}$ is the $(K-1)$-dimensional probability simplex $\{(p_k)_{k \in [K]} \mid \sum_{k=1}^K p_k = 1, p_k \in [0,1] \text{ for } k = 1, \ldots, K\}$. Moreover, if the CPD $(\Pr(Y = y|\boldsymbol{X} = \boldsymbol{x}))_{y \in [K]}$ is unimodal at any $\boldsymbol{x}$ in whole the domain $\mathbb{R}^d$ or in its sub-domain $\mathcal{X} \subseteq \mathbb{R}^d$ with a large probability $\Pr(\boldsymbol{X} \in \mathcal{X})$, we say that the data is unimodal or almost-unimodal.*

This paper presents a qualitative discussion on the representation ability of various statistical OR models. It aims to clarify whether each model is suitable for representing data that follow a distribution with various structural properties that we consider to be expectable in unimodal ordinal data. As a preparation for that discussion, we further introduce notions that are related to structural properties of the data distribution.

We first define decay rates of a PMF as the rate at which the probability value of that PMF decays from the mode toward both ends (1 and $K$):

**Definition 2.** *For a PMF $\boldsymbol{p} = (p_k)_{k \in [K]} \in \bar{\Delta}_{K-1}$ having a mode $m \in [K]$, we call $\frac{p_k}{p_{k+1}}$ for $k = 1, \ldots, m-1$ (if $m \neq 1$) and $\frac{p_k}{p_{k-1}}$ for $k = m+1, \ldots, K$ (if $m \neq K$) the decay rates (DRs) of $\boldsymbol{p}$, where $\bar{\Delta}_{K-1} \coloneqq \{(p_k)_{k \in [K]} \mid \sum_{k=1}^K p_k = 1, p_k \in (0,1) \text{ for } k = 1, \ldots, K\}$.*

Note that we here introduced $\bar{\Delta}_{K-1}$ to avoid the situation where a probability value becomes exactly 0 and technical problems associated with such a situation. It will appear several times in the following discussion.

Commonly, scale of a PMF refers to the degree of spread of that PMF. The following definition gives a numerical characterization of the scale used in this paper and introduces related notions.

---

1 $\arg\min$ and $\arg\max$ may return a set, and so $\min$ is applied to convert it to a point and simplify the discussion.

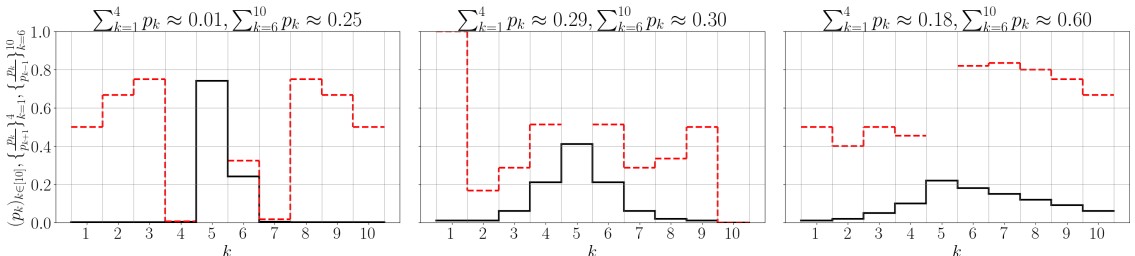

Figure 2: Instances of the unimodal 10-dimensional PMFs $(p_k)_{k \in [10]}$, in which the mode is 5, in black solid line, and their DRs $\{\frac{p_k}{p_{k+1}}\}_{k=1}^4$ and $\{\frac{p_k}{p_{k-1}}\}_{k=6}^{10}$ in red dotted line.

**Definition 3.** *For a PMF $\boldsymbol{p} = (p_k)_{k \in [K]} \in \Delta_{K-1}$ having a mode $m \in [K]$, we call $\sum_{k=1}^{m-1} p_k + \sum_{k=m+1}^{K} p_k = 1 - p_m$ the scale of $\boldsymbol{p}$, where we define the summation notation $\sum$ such that $\sum_{k=i}^{j} f_k = 0$ as far as $i > j$ regardless of $\{f_k\}$. If the scale of the CPD $\boldsymbol{p}(\boldsymbol{x}) = (\Pr(Y = y | \boldsymbol{X} = \boldsymbol{x}))_{y \in [K]}$ is similar or dissimilar over whole the domain $\mathbb{R}^d$ of the explanatory variables $\boldsymbol{X}$, we say that the data is homoscedastic or heteroscedastic. In particular, we refer to heteroscedastic data as mode-wise heteroscedastic or overall heteroscedastic data, if the scale of their $\boldsymbol{p}(\boldsymbol{x})$ is similar or can be dissimilar in a domain where the conditional mode $M(\boldsymbol{p}(\boldsymbol{x}))$ is the same.*

Commonly, skewness of a distribution refers to the asymmetricity of that distribution (around its mode). The following definition gives a numerical characterization of the skewness of the PMF used in this paper and introduces related notions.

**Definition 4.** *For a PMF $\boldsymbol{p} = (p_k)_{k \in [K]} \in \Delta_{K-1}$ having a mode $m \in [K]$, we call $\sum_{k=1}^{m-1} p_k - \sum_{k=m+1}^{K} p_k$ the skewness of $\boldsymbol{p}$, and we say that $\boldsymbol{p}$ is skew or less skew when the absolute value of its skewness (absolute skewness) is large or small. If the absolute skewness of the CPD $\boldsymbol{p}(\boldsymbol{x}) = (\Pr(Y = y | \boldsymbol{X} = \boldsymbol{x}))_{y \in [K]}$ is large or small over whole the domain $\mathbb{R}^d$ of the explanatory variables $\boldsymbol{X}$, we say that the data is skew or less skew.*

For a unimodal PMF, its DRs are smaller than or equal to 1. The scale and skewness of PMFs are typically treated as qualitative notions, but we measure them numerically for the sake of clarity (see $\sum_{k=1}^{m-1} p_k + \sum_{k=m+1}^{K} p_k$ and $\sum_{k=1}^{m-1} p_k - \sum_{k=m+1}^{K} p_k$ in Definitions 3 and 4). The numerical measure in each definition is intended just to further clarify our qualitative discussion, and we do not see any particular importance in that choice of the measure (other measures can be used instead). The readers should understand our treatment of them by referring to the description of Definitions 2, 3, and 4 and Figure 2. The figure displays 3 instances of the unimodal PMFs and their DRs: we say that the right PMF has a larger scale than the left PMF and that the center PMF is less skew and the right PMF is skew.

## 2.2 Likelihood Models and Ordinal Regression Methods

Suppose that one has a set of observations (ordinal data) $\{(\boldsymbol{x}_i, y_i)\}_{i=1}^n$, each of which is drawn independently from an identical distribution of $(\boldsymbol{X}, Y)$. Every statistical OR method covered in this paper (except for AD tried in Section 6) assumes, as a model of the conditional probability $\Pr(Y = y | \boldsymbol{X} = \boldsymbol{x})$, a certain likelihood model $\hat{\Pr}(Y = y | \boldsymbol{X} = \boldsymbol{x}) = P(y; \boldsymbol{g}(\boldsymbol{x}))$ with a fixed part $P$ (say, the softmax function in multinomial logistic regression model) and learnable part $\boldsymbol{g} \in \mathcal{G}$ (say, a certain neural network model), where we call $P$ the link function and $\boldsymbol{g}$ the learner model in a learner class $\mathcal{G}$. Note that the distinction between $P$ and $\boldsymbol{g}$ is to aid in understanding relation between multiple models, and this paper does not emphasize a strict mathematical distinction (note that, for example, $(P(\cdot; \cdot), \mathcal{G})$ and $(P(\cdot; 2\cdot), \{\boldsymbol{g}/2 \mid \boldsymbol{g} \in \mathcal{G}\})$ yield an equivalent likelihood model).

The key notion in the discussion of this paper is the representation ability of the likelihood model. Formally, the representation ability of the likelihood model we discuss is defined as follows, based on the inclusion relation of the function space corresponding to the likelihood model:

**Definition 5.** *We say that the likelihood model based on $(P_1, \mathcal{G}_1)$ has a stronger representation ability (or is more representable) in the formal sense than that based on $(P_2, \mathcal{G}_2)$, if*

$$\{(P_1(y; \boldsymbol{g}(\cdot)))_{y \in [K]} \mid \boldsymbol{g} \in \mathcal{G}_1\} \supseteq \{(P_2(y; \boldsymbol{g}(\cdot)))_{y \in [K]} \mid \boldsymbol{g} \in \mathcal{G}_2\}. \tag{2}$$

The representation abilities of two different likelihood models may not be formally comparable according to Definition 5, but even in such cases we in this paper discuss the relationship between the representation abilities from a qualitative understanding of those likelihood models as much as possible. In the following part of the paper, we describe results obtained via such a discussion as "one likelihood model has a stronger representation ability (or is more representable) in the informal sense than the other", and distinguish them from results in the formal sense that follow Definition 5.

Additionally, we introduce the functional degree of freedom:

**Definition 6.** *The functional degree of freedom (FDF) of a vector-valued function $\boldsymbol{f}_1 : \mathbb{R}^d \to \mathbb{R}^L$ to the minimum number of functions $g_1, \ldots, g_M : \mathbb{R}^d \to \mathbb{R}$ required for $\boldsymbol{f}_1(\cdot) = \boldsymbol{f}_2(g_1(\cdot), \ldots, g_M(\cdot))$ to hold with a certain vector-valued function $\boldsymbol{f}_2 : \mathbb{R}^M \to \mathbb{R}^L$.*

In this paper, we discuss the FDF mainly for CPDs or likelihood models: For example, general CPDs $(\Pr(Y = y | \boldsymbol{X} = \cdot))_{y \in [K]}$ including those underlying unimodal data have up to $(K-1)$-FDF (or called full-FDF) (it is not $K$ since $\sum_{y=1}^{K} \Pr(Y = y | \boldsymbol{X} = \cdot) = 1$, and can be smaller than $d$ when $d < K$), while likelihood models in Figure 3 are 1-FDF. We use the FDF as one simple indicator of the representation ability of statistical likelihood models; we consider that the larger its FDF, the stronger the representation ability of the statistical model tends in the informal sense.

In this paper, we set up a statistical OR method as learning a model $\boldsymbol{g}$ from the class $\mathcal{G}$ through the maximum likelihood estimation $\max_{\boldsymbol{g} \in \mathcal{G}} \frac{1}{n} \sum_{i=1}^{n} \log P(y_i; \boldsymbol{g}(\boldsymbol{x}_i))$ (which we call a conditional probability estimation task), and then constructing a classifier under the task with the task loss $\ell$ as $f(\boldsymbol{x}) = f_\ell((P(y; \hat{\boldsymbol{g}}(\boldsymbol{x})))_{y \in [K]})$ with $f_\ell((p_k)_{k \in [K]}) := \min(\arg \min_j (\sum_{k=1}^{K} p_k \ell(j, k))_{j \in [K]})$ and an obtained model $\hat{\boldsymbol{g}}$.

Therefore, the only difference of statistical OR methods in this paper appears in the link function $P$ and learner class $\mathcal{G}$ of the likelihood model. Under these settings, generic principles based on the bias-variance tradeoff suggest that a compact likelihood model that can adequately represent the data is promising for an OR method with good generalization performance. On the other hand, we believe that subsequent discussions hold as well even if changing the loss function used in the parameter fitting procedure and the decision function: As options for the parameter fitting other than the maximum likelihood estimation, we can apply robust alternatives (Bianco & Yohai, 1996; Croux et al., 2013) to every likelihood model treated. Anonymous (2022) has developed a decision function that allows misspecification of the likelihood model.

## 3 Existing Unimodal Likelihood Models

### 3.1 Binomial Model

The (shifted) binomial distribution $(P_{\mathrm{b}}(k; p))_{k \in [K]}$ is unimodal at any $p \in [0, 1]$, where

$$P_{\mathrm{b}}(k; p) := \binom{K-1}{k-1} p^{k-1}(1-p)^{K-k} \text{ for } k \in [K], \ p \in [0, 1], \tag{3}$$

and where $\binom{k}{l} := \frac{k!}{l!(k-l)!}$ is the binomial coefficient. Inspired by the shape of the PMF of the binomial distribution, da Costa et al. (2008) considered a unimodal likelihood model based on the link function

$$P(y; u) = P_{\mathrm{b}}\left(y; \frac{1}{1+e^{-u}}\right) = \binom{K-1}{y-1}\left(\frac{1}{1+e^{-u}}\right)^{y-1}\left(\frac{1}{1+e^{u}}\right)^{K-y} \tag{4}$$

and an $\mathbb{R}$-valued learner model $g(\boldsymbol{x})$ (applied to $u$ in (4)). Thereafter, Beckham & Pal (2017) introduced the scaling factor: in other words, they proposed the link function

$$P_{\mathrm{bin}}(y; u, s) := \frac{e^{\log(P_{\mathrm{b}}(y; \frac{1}{1+e^{-u}}))/s}}{\sum_{k=1}^{K} e^{\log(P_{\mathrm{b}}(k; \frac{1}{1+e^{-u}}))/s}} \text{ for } y \in [K], \ u \in \mathbb{R}, \ s \in (0, \infty), \tag{5}$$

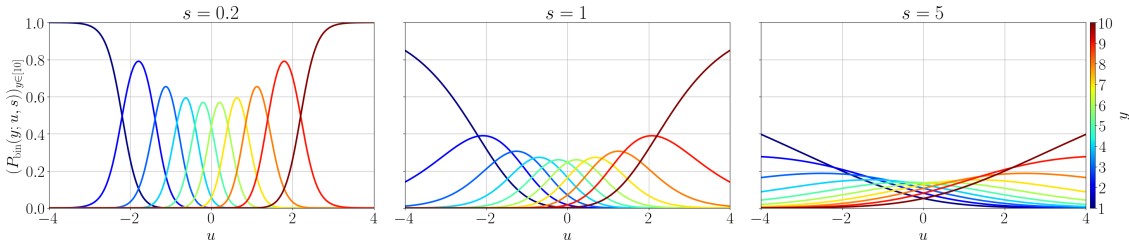

Figure 3: Instances of the BIN model with $K = 10$.[2]

and applied a $\boldsymbol{x}$-dependent $\mathbb{R}$-valued model to $u$ and a $\boldsymbol{x}$-independent positive parameter to $s$. We call this likelihood model, which consists of the link function $P_{\text{bin}}$ and learner class $\mathcal{G} = \{(g(\cdot), s) \mid g : \mathbb{R}^d \to \mathbb{R}, s \in (0, \infty)\}$, the binomial (BIN) model.

The BIN model $(P_{\text{bin}}(y; g(\boldsymbol{x}), s))_{y \in [K]}$ is parametrically constrained with the 1-FDF. The mode of the binomial distribution is

$$M((P_{\text{b}}(k; p))_{k \in [K]}) = \min(\{\lceil Kp \rceil, \lfloor Kp \rfloor + 1\} \cap [K]), \tag{6}$$

where $\lceil \cdot \rceil$ and $\lfloor \cdot \rfloor$ are the ceiling and floor functions, and hence $\max\{1, Kp\} \le M((P_{\text{b}}(k; p))_{k \in [K]}) \le \min\{Kp + 1, K\}$. This property and simple calculations can show the mode, unimodality, and DRs of the BIN model:

**Theorem 1.** *It holds that*

(i) $\frac{P_{\text{bin}}(y+1; u, s)}{P_{\text{bin}}(y; u, s)} = \left(\frac{(K-y)p}{y(1-p)}\right)^{1/s}$ *with* $p = \frac{1}{1+e^{-u}}$ *for all* $y \in [K-1]$, $u \in \mathbb{R}$, $s \in (0, \infty)$.

*Then, for any* $u \in \mathbb{R}$ *and* $s \in (0, \infty)$*, it holds that, for* $m = M((P_{\text{bin}}(y; u, s))_{y \in [K]})$ *that is* (6) *with* $p = \frac{1}{1+e^{-u}}$,

(ii) $P_{\text{bin}}(1; u, s) \le \cdots \le P_{\text{bin}}(m; u, s)$ *if* $m \ne 1$*, and* $P_{\text{bin}}(m; u, s) \ge \cdots \ge P_{\text{bin}}(K; u, s)$ *if* $m \ne K$,

(iii) $\frac{P_{\text{bin}}(1; u, s)}{P_{\text{bin}}(2; u, s)} \le \cdots \le \frac{P_{\text{bin}}(m-1; u, s)}{P_{\text{bin}}(m; u, s)}$ $(\le 1)$ *if* $m \ne 1$*, and* $(1 \ge) \frac{P_{\text{bin}}(m+1; u, s)}{P_{\text{bin}}(m; u, s)} \ge \cdots \ge \frac{P_{\text{bin}}(K; u, s)}{P_{\text{bin}}(K-1; u, s)}$ *if* $m \ne K$.

First note that Theorem 1 (ii) can be seen as a direct corollary of Theorem 1 (iii). Theorem 1 (ii) implies that the BIN model is guaranteed to be unimodal. We call the constraint on the sequence of DRs like Theorem 1 (iii) as the DRs' unimodality (constraint), considering that $(\frac{p_1}{p_2}, \ldots, \frac{p_{m-1}}{p_m}, \frac{p_{m+1}}{p_m}, \ldots, \frac{p_K}{p_{K-1}})$ is unimodal if a PMF $(p_k)_{k \in [K]}$ having a mode $m$ satisfies that constraint. One can find that, owing to the DRs' unimodality constraint, the BIN model cannot exactly represent unimodal MPFs in Figure 2. In addition to the DRs' unimodality constraint, the BIN model $(P_{\text{bin}}(y; g(\boldsymbol{x}), s))_{y \in [K]}$ is always less skew especially when its mode is close to the labels' intermediate value $\frac{1+K}{2}$ (recall that the mode (6) and median, $\lceil (K-1)p \rceil + 1$ or $\lfloor (K-1)p \rfloor + 1$, of the binomial distribution $(P_{\text{b}}(k; p))_{k \in [K]}$ are close (Kaas & Buhrman, 1980)), and tends homoscedastic (see Figure 3; there, the scale $1 - \max_y (P_{\text{bin}}(y; u, s))_{y \in [10]}$ for each $s$ is similar for many $u$ except at both ends). We will relax the restriction of the representation ability regarding the scale of the BIN model in Section 5.2, but that relaxed model cannot avoid the restriction regarding the DRs and skewness.

### 3.2 Poisson Model

The (shifted) Poisson distribution $(P_{\text{p}}(k; \lambda))_{k \in \mathbb{N}}$ is unimodal at any $\lambda > 0$, where

$$P_{\text{p}}(k; \lambda) \coloneqq \frac{\lambda^{k-1} e^{-\lambda}}{(k-1)!} \text{ for } k \in \mathbb{N}, \ \lambda \in (0, \infty). \tag{7}$$

da Costa et al. (2008) truncated (7) within $[K]$ and normalize it to develop a unimodal likelihood model: $\hat{\text{Pr}}(Y = y | \boldsymbol{X} = \boldsymbol{x}) = P_{\text{p}}(y; g(\boldsymbol{x})) / \sum_{k=1}^{K} P_{\text{p}}(k; g(\boldsymbol{x}))$ with a positive-valued learner model $g(\boldsymbol{x})$. As in the case for

---

[2]The display style in Figure 2 is like showing a PMF created by cutting the curves in Figure 3 out at each $u$.

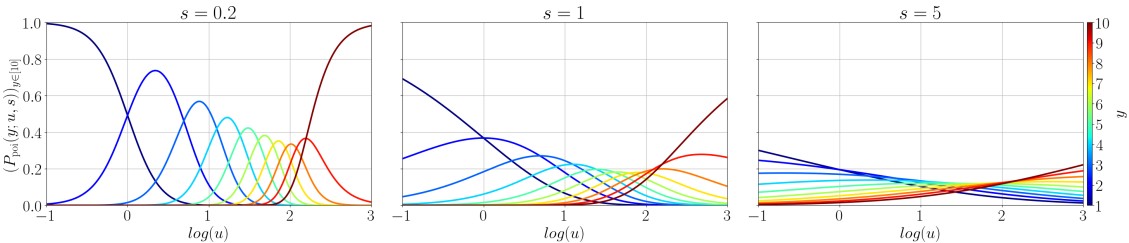

Figure 4: Instances of the POI model with $K = 10$.

the BIN model, Beckham & Pal (2017) developed a scaled link function

$$P_{\text{poi}}(y; u, s) := \frac{e^{v_y/s}}{\sum_{k=1}^{K} e^{v_k/s}} \text{ for } y \in [K], \ u, s \in (0, \infty), \text{ with } v_k = (k-1)\log(u) - \log((k-1)!) \text{ for } k \in [K], \quad (8)$$

which is a generalization of $P_{\text{poi}}(y; u, 1)$ of (da Costa et al., 2008).[3] They applied a $\boldsymbol{x}$-dependent positive-valued learner model $g(\boldsymbol{x})$ to $u$ and a $\boldsymbol{x}$-independent positive parameter to $s$ (i.e., $\mathcal{G} = \{(g(\cdot), s) \mid g : \mathbb{R}^d \to (0, \infty), s \in (0, \infty)\}$); we call this likelihood model the Poisson (POI) model.

The POI model $(P_{\text{poi}}(y; g(\boldsymbol{x}), s))_{y \in [K]}$ is also parametrically constrained with the 1-FDF. We find that the POI model is a special instance of PO-ORD-ACL and PO-ACL models discussed later (in Sections 4.1, 5.1, and 5.2) and has a weaker representation ability in the formal sense; see Theorem 15. The POI model $(P_{\text{poi}}(y; g(\boldsymbol{x}), s))_{y \in [K]}$ has a DRs' unimodality constraint similar to the BIN model. Also, it always has mode-wise heteroscedasticity especially when $s$ is small and $K$ is large: its scale tends small and large respectively when $M((P_{\text{poi}}(y; g(\boldsymbol{x}), s))_{y \in [K]})$ is close to 1, $K$, or smaller side of $\{2, \ldots, K-1\}$ and larger side; see Figure 4 with $s = 0.2$. Therefore, it is not suitable for representing homoscedastic or overall heteroscedastic data.

## 4 Novel Unimodal Likelihood Models

### 4.1 Ordered Adjacent Categories Logit Model

The previous BIN and POI models are ensured to be unimodal, but have just 1-FDF and may be too weakly representable for some data. We thus developed more strongly representable unimodal likelihood models that have up to $(K-1)$-FDF. This section describes ordered adjacent categories logit (ORD-ACL) model developed by modifying existing ACL model that does not have the unimodality guarantee.

The (naïve) ACL model is designed to model the relationship between the underlying conditional probabilities of the adjacent categories, $\frac{\Pr(Y=y|\boldsymbol{X}=\boldsymbol{x})}{\Pr(Y\in\{y,y+1\}|\boldsymbol{X}=\boldsymbol{x})}$, as

$$\frac{\hat{\Pr}(Y = y | \boldsymbol{X} = \boldsymbol{x})}{\hat{\Pr}(Y \in \{y, y+1\} | \boldsymbol{X} = \boldsymbol{x})} = \frac{1}{1 + e^{-g_y(\boldsymbol{x})}} \text{ for } y \in [K-1] \quad (9)$$

with an $\mathbb{R}^{(K-1)}$-valued learner model $\boldsymbol{g}$; see Simon (1974); Andrich (1978); Goodman (1979); Masters (1982), and Agresti (2010, Section 4.1). Considering the normalization condition $\sum_{y=1}^{K} \hat{\Pr}(Y = y | \boldsymbol{X} = \boldsymbol{x}) = 1$, it can be seen that the ACL model depends on the ACL link function

$$P_{\text{acl}}(y; \boldsymbol{u}) := \frac{\prod_{l=1}^{y-1} e^{-u_l}}{\sum_{k=1}^{K} \prod_{l=1}^{k-1} e^{-u_l}} = \frac{e^{-\sum_{l=1}^{y-1} u_l}}{\sum_{k=1}^{K} e^{-\sum_{l=1}^{k-1} u_l}} \text{ for } y \in [K], \ \boldsymbol{u} \in \mathbb{R}^{K-1}, \quad (10)$$

together with the learner class $\mathcal{G} = \{\boldsymbol{g} : \mathbb{R}^d \to \mathbb{R}^{K-1}\}$, where we define the product notation $\prod$ such that $\prod_{k=i}^{j} f_k = 1$ as far as $i > j$ regardless of $\{f_k\}$.

The ACL model can represent arbitrary CPD in the following sense:

---

[3]In (Beckham & Pal, 2017), $v_k$ in (8) is defined with minus $\boldsymbol{u}$, but which is irrelevant after the softmax transformation.

**Theorem 2.** *It holds that*

$$\{(P_{\mathrm{acl}}(y; \boldsymbol{g}(\cdot)))_{y \in [K]} \mid \boldsymbol{g} : \mathbb{R}^d \to \mathbb{R}^{K-1}\} = \{\boldsymbol{p} : \mathbb{R}^d \to \bar{\Delta}_{K-1}\}. \tag{11}$$

Theorem 2 implies that the naïve ACL model does not have the unimodality guarantee, while we analyzed properties of the ACL link function and then found a simple condition for its learner class under which a likelihood model based on the ACL link function gets the unimodality guarantee:

**Theorem 3.** *It holds that*

(i) $\frac{P_{\mathrm{acl}}(y+1; \boldsymbol{u})}{P_{\mathrm{acl}}(y; \boldsymbol{u})} = e^{-u_y}$ *for all* $y \in [K-1]$, $\boldsymbol{u} \in \mathbb{R}^{K-1}$.

*If* $\boldsymbol{u} \in \mathbb{R}^{K-1}$ *satisfies* $u_0(:= -\infty) \le \cdots \le u_{m-1} \le 0 \le u_m \le \cdots \le u_K(:= \infty)$ *for some* $m \in [K]$, *it holds that*

(ii) $P_{\mathrm{acl}}(1; \boldsymbol{u}) \le \cdots \le P_{\mathrm{acl}}(m; \boldsymbol{u})$ *if* $m \ne 1$, *and* $P_{\mathrm{acl}}(m; \boldsymbol{u}) \ge \cdots \ge P_{\mathrm{acl}}(K; \boldsymbol{u})$ *if* $m \ne K$,

(iii) $\frac{P_{\mathrm{acl}}(1; \boldsymbol{u})}{P_{\mathrm{acl}}(2; \boldsymbol{u})} \le \cdots \le \frac{P_{\mathrm{acl}}(m-1; \boldsymbol{u})}{P_{\mathrm{acl}}(m; \boldsymbol{u})}$ $(\le 1)$ *if* $m \ne 1$, *and* $(1 \ge) \frac{P_{\mathrm{acl}}(m+1; \boldsymbol{u})}{P_{\mathrm{acl}}(m; \boldsymbol{u})} \ge \cdots \ge \frac{P_{\mathrm{acl}}(K; \boldsymbol{u})}{P_{\mathrm{acl}}(K-1; \boldsymbol{u})}$ *if* $m \ne K$.

On the basis of the unimodality guarantee stated in Theorem 3 (ii), we propose the ORD-ACL model $(P_{\mathrm{acl}}(y; \acute{\boldsymbol{g}}(\boldsymbol{x})))_{y \in [K]}$ applying the ACL link function (10) and ordered learner model $\acute{\boldsymbol{g}}(\boldsymbol{x})$ satisfying that $\acute{g}_1(\boldsymbol{x}) \le \cdots \le \acute{g}_{K-1}(\boldsymbol{x})$ for any $\boldsymbol{x} \in \mathbb{R}^d$. Here, the ordered learner model $\acute{\boldsymbol{g}}$ can be implemented as

$$\acute{g}_k(\boldsymbol{x}) = \begin{cases} g_1(\boldsymbol{x}), & \text{for } k = 1, \\ \acute{g}_{k-1}(\boldsymbol{x}) + \rho(g_k(\boldsymbol{x})), & \text{for } k = 2, \ldots, K-1, \end{cases} \tag{12}$$

with another model $\boldsymbol{g} : \mathbb{R}^d \to \mathbb{R}^{K-1}$ and a fixed function $\rho$ that satisfies

$$\{\rho(u) \mid u \in \mathbb{R}\} = [0, \infty) \tag{13}$$

such as $\rho_{\mathrm{sq}}(u) := u^2$ (we denote the procedure (12) as $\acute{\boldsymbol{g}} = \rho[\boldsymbol{g}]$).[4]

The ORD-ACL model can represent arbitrary unimodal CPD that has the DRs' unimodality constraint:

**Theorem 4.** *For any function* $\rho$ *satisfying* (13), *it holds that*

$$\{(P_{\mathrm{acl}}(y; \rho[\boldsymbol{g}(\cdot)]))_{y \in [K]} \mid \boldsymbol{g} : \mathbb{R}^d \to \mathbb{R}^{K-1}\}$$
$$= \{\boldsymbol{p} : \mathbb{R}^d \to \bar{\Delta}_{K-1} \mid (\frac{p_1(\boldsymbol{x})}{p_2(\boldsymbol{x})}, \ldots, \frac{p_{M(\boldsymbol{p}(\boldsymbol{x}))-1}(\boldsymbol{x})}{p_{M(\boldsymbol{p}(\boldsymbol{x}))}(\boldsymbol{x})}, \frac{p_{M(\boldsymbol{p}(\boldsymbol{x}))+1}(\boldsymbol{x})}{p_{M(\boldsymbol{p}(\boldsymbol{x}))}(\boldsymbol{x})}, \ldots, \frac{p_K(\boldsymbol{x})}{p_{K-1}(\boldsymbol{x})}) \text{ is unimodal for any } \boldsymbol{x} \in \mathbb{R}^d\}. \tag{14}$$

It will be a good aspect in the modeling of the ordinal data assumed to be unimodal that the ORD-ACL model has the unimodality guarantee. In contrast, the ORD-ACL model incidentally imposes a constraint on the DRs; even so it is important that the ORD-ACL model has a stronger representation ability in the formal sense than the BIN and POI models, which also have DRs' unimodality constraint.

### 4.2 V-Shaped Stereotype Logit Model

We next describe V-shaped stereotype logit (VS-SL) model, which is unimodal, full-FDF, more representable than the ORD-ACL model in the formal sense (see Theorem 10), and developed by modifying existing SL model (also called multinomial logistic regression models).

The SL model (refer to Anderson (1984) and Agresti (2010, Section 4.3)) attempts to model the conditional probabilities of multiple categories paired with a certain fixed (stereotype) category, $\frac{\mathrm{Pr}(Y=1 \mid \boldsymbol{X}=\boldsymbol{x})}{\mathrm{Pr}(Y \in \{1,y\} \mid \boldsymbol{X}=\boldsymbol{x})}$, as

$$\frac{\hat{\mathrm{Pr}}(Y = 1 \mid \boldsymbol{X} = \boldsymbol{x})}{\hat{\mathrm{Pr}}(Y \in \{1, y\} \mid \boldsymbol{X} = \boldsymbol{x})} = \frac{1}{1 + e^{-g_y(\boldsymbol{x})}} \text{ for } y \in [K] \tag{15}$$

---

[4]If a non-zero alternative $\rho_{\mathrm{exp}}(u) := e^u$ is used as the function $\rho$, inequalities in the ordering condition of $\acute{\boldsymbol{g}}$ and the unimodality condition reduce to the strict inequalities. Using such a $\rho$ function does not cause serious computational problems.

with an $\mathbb{R}^K$-valued learner model $\boldsymbol{g}$ such that $g_1(\boldsymbol{x}) = 0$ for any $\boldsymbol{x} \in \mathbb{R}^d$. Considering the normalization condition, we introduce the SL link function as

$$P_{\mathrm{sl}}(y; \boldsymbol{u}) := \frac{e^{-u_y}}{\sum_{k=1}^K e^{-u_k}} \text{ for } y \in [K], \ \boldsymbol{u} \in \mathbb{R}^K. \tag{16}$$

The SL model is based on the SL link function $P_{\mathrm{sl}}$ and learner class $\mathcal{G} = \{\boldsymbol{g} : \mathbb{R}^d \to \mathbb{R}^K \mid g_1(\cdot) = 0\}$.

The SL model can represent arbitrary CPD:

**Theorem 5.** *It holds that*

$$\{(P_{\mathrm{sl}}(y; \boldsymbol{g}(\cdot)))_{y \in [K]} \mid \boldsymbol{g} : \mathbb{R}^d \to \mathbb{R}^K, g_1(\cdot) = 0\} = \{(P_{\mathrm{sl}}(y; \boldsymbol{g}(\cdot)))_{y \in [K]} \mid \boldsymbol{g} : \mathbb{R}^d \to \mathbb{R}^K\} = \{\boldsymbol{p} : \mathbb{R}^d \to \bar{\Delta}_{K-1}\}. \tag{17}$$

The ACL and SL link functions have only minor differences in parameterization, but looking at the SL link function will reveal another simple way to create a unimodal likelihood model as follow:

**Theorem 6.** *If $\boldsymbol{u} \in \mathbb{R}^K$ satisfies that $u_1 \geq \cdots \geq u_m$ if $m \neq 1$ and $u_m \leq \cdots \leq u_K$ if $m \neq K$ for some $m \in [K]$, it holds that $P_{\mathrm{sl}}(1; \boldsymbol{u}) \leq \cdots \leq P_{\mathrm{sl}}(m; \boldsymbol{u})$ if $m \neq 1$ and $P_{\mathrm{sl}}(m; \boldsymbol{u}) \geq \cdots \geq P_{\mathrm{sl}}(K; \boldsymbol{u})$ if $m \neq K$.*

On the ground of this theorem, we proposed VS-SL model $(P_{\mathrm{sl}}(y; \check{\boldsymbol{g}}(\boldsymbol{x})))_{y \in [K]}$ based on the SL link function $P_{\mathrm{sl}}$ and a V-shaped learner model $\check{\boldsymbol{g}} = \tau(\acute{\boldsymbol{g}})$ (described below) with $\acute{\boldsymbol{g}} = \rho[\boldsymbol{g}]$ and another $\mathbb{R}^K$-valued learner model $\boldsymbol{g}$. Namely, the VS-SL model applies the learner class $\mathcal{G} = \{\tau(\rho[\boldsymbol{g}]) \mid \boldsymbol{g} : \mathbb{R}^d \to \mathbb{R}^K\}$ for the link function $P_{\mathrm{sl}}$. First, $\rho$ transforms an arbitrary model $\boldsymbol{g}$ to an ordered model $\acute{\boldsymbol{g}}$ (so $\acute{g}_1(\boldsymbol{x}) \leq \cdots \leq \acute{g}_K(\boldsymbol{x})$ for any $\boldsymbol{x} \in \mathbb{R}^d$). Next, $\tau$ transforms an ordered model $\acute{\boldsymbol{g}}$ to a V-shaped model $\tau(\acute{\boldsymbol{g}})$. Here, the notation $\tau(\boldsymbol{u})$ for an $\mathbb{R}^K$-valued object $\boldsymbol{u}$ implies the element-wise application $(\tau \circ u_k)_{k \in [K]}$, and the function $\tau$ is supposed to satisfy

$$\begin{aligned} &\tau(u) \text{ is non-increasing in } u < 0 \text{ and non-decreasing in } u > 0, \\ &\text{and } \{\tau(u) \mid u \leq 0\} = \{\tau(u) \mid u \geq 0\} = [\tau(0), \infty), \end{aligned} \tag{18}$$

such as $\tau_{\mathrm{abs}}(u) := |u|$ and $\tau_{\mathrm{sq}}(u) := u^2$. If $\tau = \tau_{\mathrm{abs}}, \tau_{\mathrm{sq}}$, it holds that $\check{g}_1(\boldsymbol{x}) \geq \cdots \geq \check{g}_{m(\boldsymbol{x})}(\boldsymbol{x})$ and $\check{g}_{m(\boldsymbol{x})}(\boldsymbol{x}) \leq \cdots \leq \check{g}_K(\boldsymbol{x})$ with $m(\boldsymbol{x}) = \min(\arg\min_k(|\acute{g}_k(\boldsymbol{x})|)_{k \in [K]})$, as required in Theorem 6. Finally, the SL link function transforms $\check{\boldsymbol{g}}$ to a unimodal likelihood model.

The VS-SL model $(P_{\mathrm{sl}}(y; \check{\boldsymbol{g}}(\boldsymbol{x})))_{y \in [K]}$ is ensured to be unimodal and further can represent arbitrary unimodal CPD:

**Theorem 7.** *For any functions $\rho$ satisfying* (13) *and $\tau$ satisfying* (18)*, it holds that*

$$\{(P_{\mathrm{sl}}(y; \tau(\rho[\boldsymbol{g}(\cdot)])))_{y \in [K]} \mid \boldsymbol{g} : \mathbb{R}^d \to \mathbb{R}^K\} = \{\boldsymbol{p} : \mathbb{R}^d \to \bar{\Delta}_{K-1} \mid \boldsymbol{p}(\boldsymbol{x}) \text{ is unimodal for any } \boldsymbol{x} \in \mathbb{R}^d\}. \tag{19}$$

Accordingly, we propose the VS-SL model as the most representable one among the class of the unimodal likelihood models.

# 5 Variant Models

## 5.1 Proportional-Odds Models

The ACL model $(P_{\mathrm{acl}}(y; \boldsymbol{g}(\boldsymbol{x})))_{y \in [K]}$, ORD-ACL model $(P_{\mathrm{acl}}(y; \rho[\boldsymbol{g}(\boldsymbol{x})]))_{y \in [K]}$, SL model $(P_{\mathrm{sl}}(y; \boldsymbol{g}(\boldsymbol{x})))_{y \in [K]}$, and VS-SL model $(P_{\mathrm{sl}}(y; \tau(\rho[\boldsymbol{g}(\boldsymbol{x})])))_{y \in [K]}$ have up to full-FDF and may be too flexible and difficult to learn for some data especially when the sample size is not large enough. In the statistical research (e.g., see McCullagh (1980)), the proportional-odds (PO) constraint that constrains the FDF of the likelihood model to 1 is often applied to avoid such a trouble. We call $(P_{\mathrm{acl}}(y; \boldsymbol{b} - g(\boldsymbol{x}) \cdot \mathbf{1}))_{y \in [K]}$ the PO-ACL model, $(P_{\mathrm{acl}}(y; \rho[\boldsymbol{b}] - g(\boldsymbol{x}) \cdot \mathbf{1}))_{y \in [K]}$ the PO-ORD-ACL model, and $(P_{\mathrm{sl}}(y; \tau(\rho[\boldsymbol{b}] - g(\boldsymbol{x}) \cdot \mathbf{1})))_{y \in [K]}$ the PO-VS-SL model, where $\boldsymbol{b}$ and $\mathbf{1}$ are learnable parameter and all-1 vector with appropriate dimension ($(K-1)$ for ACL and ORD-ACL models, or $K$ for VS-SL models) and $g$ is an $\mathbb{R}$-valued learner model.[5]

---

[5]The PO-SL model $P_{\mathrm{sl}}(y; \boldsymbol{b} - g(\boldsymbol{x}) \cdot \mathbf{1}) = e^{-b_y} / \sum_{k=1}^K e^{-b_k}$ is constant regarding $\boldsymbol{x}$ and may be weakly representable. Therefore, Anderson (1984) considered to use a learner model $\boldsymbol{g}(\boldsymbol{x}) = \boldsymbol{a} - g(\boldsymbol{x}) \cdot \boldsymbol{c}$ with learnable parameters $\boldsymbol{a}, \boldsymbol{c}$ ($\in \mathbb{R}^K$ here) s.t. $a_1 = c_1 = 0$

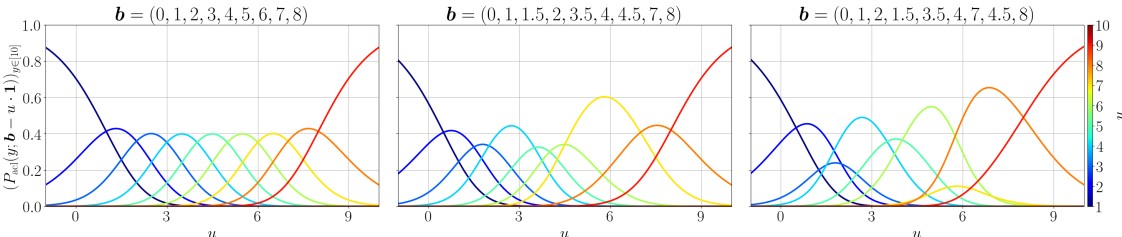

Figure 5: Instances of the PO-ACL model with $K = 10$.

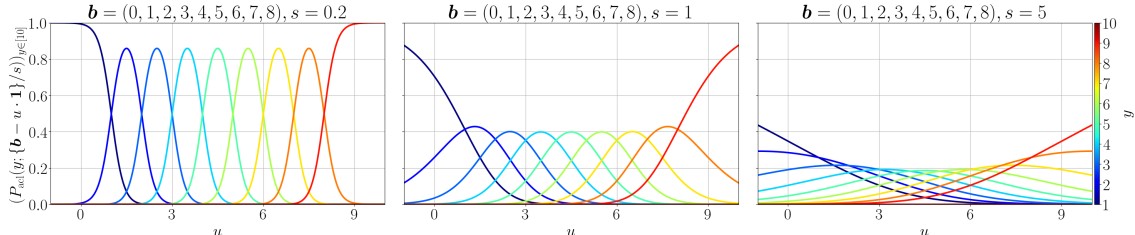

Figure 6: Instances of the OH-ACL model with $K = 10$.

Recall the POI model reviewed in Section 3.2. The equation

$$P_{\mathrm{poi}}(y; u, s) = P_{\mathrm{acl}}(y; \boldsymbol{b} - v \cdot \boldsymbol{1}) \text{ for all } y \in [K], \text{ with } \boldsymbol{b} = (\log(k)/s)_{k \in [K-1]}, \; v = \log(u)/s, \tag{20}$$

which is formalized in Theorem 15, shows that the PO-ACL and PO-ORD-ACL models are more representable in the formal sense than the POI model. The PO-ACL and PO-ORD-ACL models can further represent homoscedastic data and more various mode-wise heteroscedastic data because they can adjust the parameter $\boldsymbol{b}$ or $\rho[\boldsymbol{b}]$ according to the model fit to the data, compared with the POI model; compare Figure 4 for the POI model and Figure 5 for the PO-ACL and PO-ORD-ACL models.

## 5.2 Overall Heteroscedastic Models

The BIN, POI, and PO models have just 1-FDF. The strong constraint of the representation ability of these models can be interpreted as incidentally assuming the homoscedasticity or mode-wise heteroscedasticity. We in this section describe their overall homoscedastic (OH) extension, OH models.

McCullagh (1980) studied an OH extension of a certain PO model. According to his idea, we propose the OH-ACL model $(P_{\mathrm{acl}}(y; \{\boldsymbol{b} - g(\boldsymbol{x}) \cdot \boldsymbol{1}\}/s(\boldsymbol{x})))_{y \in [K]}$, OH-ORD-ACL model $(P_{\mathrm{acl}}(y; \{\rho[\boldsymbol{b}] - g(\boldsymbol{x}) \cdot \boldsymbol{1}\}/s(\boldsymbol{x})))_{y \in [K]}$, and OH-VS-SL model $(P_{\mathrm{sl}}(y; \tau(\rho[\boldsymbol{b}] - g(\boldsymbol{x}) \cdot \boldsymbol{1})/s(\boldsymbol{x})))_{y \in [K]}$, where $\boldsymbol{b}$ and $\boldsymbol{1}$ are learnable parameter and all-1 vector with appropriate dimension, and $g$ and $s$ are $\mathbb{R}$- and positive-valued learner models. The positive-valued scale model $s$ can be implemented as $s(\boldsymbol{x}) = r + \rho(t(\boldsymbol{x}))$ with a constant $r > 0$ for avoiding zero-division, fixed non-negative function $\rho$, and another $\mathbb{R}$-valued learner $t$.

The significance of OH models is that their scaling factor can vary depending on $\boldsymbol{x}$. Extending previous BIN and POI models by Beckham & Pal (2017), we also introduce the OH-BIN model $(P_{\mathrm{bin}}(y; g(\boldsymbol{x}), s(\boldsymbol{x})))_{y \in [K]}$ and OH-POI model $(P_{\mathrm{poi}}(y; g(\boldsymbol{x}), s(\boldsymbol{x})))_{y \in [K]}$. For the implementation $s(\boldsymbol{x}) = r + \rho(t(\boldsymbol{x}))$ of the scaling factor for these models, the use of a small $r > 0$ enables to represent a wider class distribution. Note that the OH-POI model is a special instance of the OH-ACL and OH-ORD-ACL models in the formal sense, as Theorem 15 shows.

---

and $\mathbb{R}$-valued learner model $g : \mathbb{R}^d \to \mathbb{R}$ (we call $g$ a rotatable-odds (RO) learner model) for the SL link function; we call this model the RO-SL model. We can also consider novel RO-ACL model: it is more representable than the PO-ACL (and PO-ORD-ACL) model, but has no unimodality guarantee since the RO learner model $g(\boldsymbol{x}) = \boldsymbol{a} - g(\boldsymbol{x}) \cdot \boldsymbol{c}$ does not necessarily satisfy the ordering condition $g_1(\boldsymbol{x}) \leq \cdots \leq g_{K-1}(\boldsymbol{x})$ (depending on $\boldsymbol{c}$).

These OH models can represent overall heteroscedastic data and are more representable in the formal sense than their corresponding 1-FDF model (see Theorem 14), but their representation ability is parametrically constrained with the 2-FDF; compare Figures 3, 4, 5, and 6.

### 5.3 Combination-with-Uniform Models

When a likelihood model $(\hat{\Pr}(Y = y | \boldsymbol{X} = \boldsymbol{x}))_{y \in [K]}$ is unimodal, the combination-with-uniform (CU) likelihood model, $q(\boldsymbol{x}) \cdot (\frac{1}{K})_{k \in [K]} + \{1 - q(\boldsymbol{x})\} \cdot (\hat{\Pr}(Y = y | \boldsymbol{X} = \boldsymbol{x}))_{y \in [K]}$ with e.g., $q(\boldsymbol{x}) = \frac{1}{1 + e^{-t(\boldsymbol{x})}}$ for an $\mathbb{R}$-valued learner $t$, is also unimodal. For example, Piccolo (2003); Iannario & Piccolo (2011) studied a combination of uniform and binomial model. The CU extension of low-FDF (like PO and OH) models can increase their FDF by 1.

## 6 Experiments

### 6.1 Experimental Purposes

From the bias-variance tradeoff, it can be expected that compact likelihood models that can adequately represent the data will yield better generalization performance in the conditional probability estimation task, and accordingly that OR methods based on such likelihood models will yield better generalization performance in the OR task. We performed numerical experiments in order to verify whether the proposed likelihood models, developed based on this working hypothesis, can provide better performances than previous unimodal likelihood models with a weak representation ability and popular statistical OR models with no unimodality guarantee.

### 6.2 Experimental Settings

We selected 21 real-world datasets of those used in experiments by the previous OR study (Gutierrez et al., 2015) with the total sample size $n_{\text{tot}}$ that is 1000 or more, and used them for our numerical experiments.[6] AB5, ..., CE5' (resp. AB10, ..., CE10') are datasets generated by discretizing a real-valued target of datasets, which are often used to benchmark regression methods, by 5 (resp. 10) different bins with equal proportions. SW, ..., CA originally have a categorical target, and the authors of (Gutierrez et al., 2015) judged that their targets have a natural ordinal relation.

Table 1 shows the dataset name, dataset properties $n_{\text{tot}}$, $d$, and $K$, and mean and standard deviation (STD) of 100 test MUs and test DRs' MUs. The mean unimodality (MU) is a numerical criterion to evaluate the unimodality of the conditional probability distribution of the data, and it is defined, for a likelihood model $\hat{\Pr}(Y = \cdot | \boldsymbol{X} = \cdot)$ and $n$ used data points, as $\frac{1}{n} \sum_{i=1}^{n} \mathbb{1}\{(\hat{\Pr}(Y = y | \boldsymbol{X} = \boldsymbol{x}_i))_{y \in [K]} \text{ is unimodal.}\}$. As that likelihood model, we used the test SL model under the task-P (described below) that can represent any conditional probability distribution. We trained a likelihood model with a training sample of size $n_{\text{tra}} = 800$, and evaluated the MU with an obtained likelihood model and a remaining test sample of size $n_{\text{tes}} = n_{\text{tot}} - n_{\text{tra}}$. We repeated this procedure 100 trials with a randomly-set different sample setting and initial parameters of the likelihood model to obtain 100 test MUs. We also give a relative reference value in Table 2: It shows a ratio that $10^6$ samples of the PMF uniformly and randomly drawn from the probability simplex in $\mathbb{R}^K$ were unimodal for $K = 3, \ldots, 10$. Comparing the test MU in Table 1 and the relative reference value of the same $K$ in Table 2, one can find that many real-world data treated as ordinal data by existing OR research tend strongly unimodal. Additionally, we introduced the DRs' MU $\frac{1}{n} \sum_{i=1}^{n} \mathbb{1}\{(\hat{\Pr}(Y = y | \boldsymbol{X} = \boldsymbol{x}_i))_{y \in [K]} \text{ satisfies the DRs' unimodality.}\}$, and took a similar procedure to that for the MU; see Tables 1 and 2. The DRs' MU for ordinal data were larger than those for uniform random PMF, but their difference appears to be relatively smaller than that for the MU.[7]

---

[6]One can get the datasets from a researchers' site (`http://www.uco.es/grupos/ayrna/orreview`) of (Gutierrez et al., 2015), or our GitHub repository (`https://github.com/yamasakiryoya/ULM`) together with our used program codes.

[7]For notions related to the scale and skewness, we introduced numerical measures in Definitions 3 and 4 just for the sake of clarity of the qualitative discussion. However, we did not introduce metrical meaning into these measures (and it is quite difficult): for example, a gap between scales 0.1 and 0.3 and a gap between scales 0.7 and 0.9 would not have the same meaning. Therefore, we could not evaluate the heteroscedasticity without misleading.

Table 1: Dataset name, dataset properties $n_{\text{tot}}$, $d$, and $K$, and mean and STD (as 'mean ± STD') of test MUs and test DRs' MUs.

| dataset name | $n_{\text{tot}}$ | $d$ | $K$ | MU | DRs' MU |
|---|---|---|---|---|---|
| AB5 (abalon5) | 4177 | 10 | 5 | .8915 ± .0662 | .4042 ± .0743 |
| BA5 (bank5) | 8192 | 8 | 5 | .9944 ± .0387 | .4293 ± .1389 |
| BA5' (bank5') | 8192 | 32 | 5 | .9887 ± .0167 | .7041 ± .1210 |
| CO5 (computer5) | 8192 | 12 | 5 | .9956 ± .0137 | .5623 ± .1041 |
| CO5' (computer5') | 8192 | 21 | 5 | 1.0000 ± .0001 | .5851 ± .1170 |
| CH5 (cal.housing5) | 20640 | 8 | 5 | .9065 ± .1027 | .3993 ± .1206 |
| CE5 (census5) | 22784 | 8 | 5 | .7887 ± .0929 | .3221 ± .1050 |
| CE5' (census5') | 22784 | 16 | 5 | .8332 ± .0736 | .3827 ± .0978 |
| AB10 (abalon10) | 4177 | 10 | 10 | .3218 ± .1315 | .0076 ± .0164 |
| BA10 (bank10) | 8192 | 8 | 10 | .8923 ± .1463 | .0225 ± .0432 |
| BA10' (bank10') | 8192 | 32 | 10 | .5101 ± .2261 | .0019 ± .0054 |

| dataset name | $n_{\text{tot}}$ | $d$ | $K$ | MU | DRs' MU |
|---|---|---|---|---|---|
| CO10 (computer10) | 8192 | 12 | 10 | .8006 ± .1431 | .0935 ± .0883 |
| CO10' (computer10') | 8192 | 21 | 10 | .8189 ± .1617 | .0517 ± .0737 |
| CH10 (cal.housing10) | 20640 | 8 | 10 | .3311 ± .1713 | .0051 ± .0095 |
| CE10 (census10) | 22784 | 8 | 10 | .2042 ± .1017 | .0008 ± .0018 |
| CE10' (census10') | 22784 | 16 | 10 | .3151 ± .1153 | .0027 ± .0053 |
| SW (SWD) | 1000 | 10 | 4 | .9993 ± .0027 | .9853 ± .0204 |
| LE (LEV) | 1000 | 4 | 5 | .9547 ± .0666 | .2208 ± .1555 |
| ER (ERA) | 1000 | 4 | 9 | .7909 ± .1045 | .2126 ± .1403 |
| WR (winequality-red) | 1599 | 11 | 6 | .9894 ± .0457 | .1415 ± .1259 |
| CA (car) | 1728 | 21 | 4 | .8735 ± .2037 | .0276 ± .0223 |

Table 2: Ratio of $10^6$ PMF samples uniformly and randomly drawn from $\Delta_{K-1}$ that satisfied the unimodality (resp. the DRs' unimodality) in the row MU (resp. in the row DRs' MU).

| $K$ | 3 | 4 | 5 | 6 | 7 | 8 | 9 | 10 |
|---|---|---|---|---|---|---|---|---|
| MU | .6666 | .3330 | .1329 | .0444 | .0125 | .0032 | .0007 | .0001 |
| DRs' MU | .6666 | .3330 | .1329 | .0276 | .0055 | .0009 | .0001 | .0000 |

We considered 4 tasks: a conditional probability estimation task (task-P), and 3 OR tasks with the zero-one task loss $\ell_{\text{zo}}$ (task-Z), absolute task loss $\ell_{\text{abs}}$ (task-A), and squared task loss $\ell_{\text{sq}}$ (task-S). Results for the task-P, -Z, -A, and -S are respectively evaluated based on the negative log likelihood (NLL), mean zero-one error (MZE), mean absolute error (MAE), and mean squared error (MSE). Note that, for a likelihood model $\hat{\Pr}(Y = \cdot | \boldsymbol{X} = \cdot)$ and $n$ used data points, the NLL is defined as $-\frac{1}{n}\sum_{i=1}^{n}\log\hat{\Pr}(Y = y_i | \boldsymbol{X} = \boldsymbol{x}_i)$, and the MZE, MAE, and MSE are respectively defined as $\frac{1}{n}\sum_{i=1}^{n}\ell(f(\boldsymbol{x}_i), y_i)$ with $f(\boldsymbol{x}) = f_\ell((\hat{\Pr}(Y = y | \boldsymbol{X} = \boldsymbol{x}))_{y \in [K]})$ for $\ell = \ell_{\text{zo}}, \ell_{\text{abs}}, \ell_{\text{sq}}$. A method that yields a smaller error value is better for the corresponding task.

We tried 12 likelihood models and OR methods based on those likelihood models: previous 1-FDF BIN and POI models; proposed 1-FDF PO-ORD-ACL and PO-VS-SL models; proposed 2-FDF OH-BIN, OH-POI, OH-ORD-ACL, and OH-VS-SL models; proposed full-FDF ORD-ACL and VS-SL models; previous full-FDF ACL and SL models with no unimodality guarantee. See Figure 1 for the relationship among the representation abilities of these models. Here, we used link functions $\rho = \rho_{\text{exp}}$ to ensure the non-negativity and for a positive learner model for POI and OH-POI models and $\tau = \tau_{\text{sq}}$ to ensure the V-shape, and a positive constant $r = 0.01$ in the scaling model $s(\boldsymbol{x}) = r + \rho(t(\boldsymbol{x}))$ for OH models. We implemented all learner models with a 4-layer fully-connected neural network model that shares weights in except for the final layer and has 100 nodes activated with the sigmoid function in addition to bias nodes in every hidden layer. Note that, for the OH models, we implemented their real-valued learner and scaling models ($g$ and $t$ in the notation in Section 5.2) with two isolated networks (each of which is described above), because their performance was significantly degraded when the real-valued learner and scaling models were implemented with a single weight-shared network. We trained a model with a training sample and Adam optimization for 1000 epochs according to the maximum likelihood estimation, and evaluated the NLL, MZE, MAE, and MSE with a remaining test sample at the end of each epoch. Then, for the task-P, -Z, -A, or -S, we adopted a model at the timing when the test NLL, MZE, MAE, or MSE got minimum as the test model under the coresponding task.

As a baseline method, we also tried a regression-based OR method (AD) (Agarwal, 2008). This method solves least absolute deviation regression task with targets $\{y_i\}$, $\min_{g:\mathbb{R}^d \to \mathbb{R}} \frac{1}{n_{\text{tra}}}\sum_{i=1}^{n_{\text{tra}}}|g(\boldsymbol{x}_i) - y_i|$ with a regression predictor $g$ implemented with a 4-layer fully-connected neural network model, and predicts a class label for $\boldsymbol{X} = \boldsymbol{x}$ by a label value closest to $g(\boldsymbol{x})$ among $[K]$. For AD, we evaluated only the MZE, MAE, and MSE.

We experimented with 6 training sample size settings $n_{\text{tra}} = 25, 50, 100, 200, 400, 800$, to see the dependence of behaviors of each method on the training sample size.

For all combinations of 21 datasets, 4 tasks, 12/13 likelihood models, and 6 training sample size settings, we repeated above-described procedure 100 trials with a randomly-set different training sample and initial parameters, to obtain 100 test errors under the corresponding task.

Table 3: A cell for a method and training sample size $n_{\text{tra}}$ in a sub-table for an error shows the total (over the 21 datasets and 11/12 other methods) number #win (resp. #lose) of times that that method wins (resp. loses) in the Mann-Whitney U-test with $p$-value 0.05 regarding the error for each pair of that method and other method (as '#win, #lose'). A method with larger #win and smaller #lose is better. In each block for $n_{\text{tra}}$ in a sub-table, the 1st, 2nd, and 3rd best results are highlighted in the red, green, and blue colors.

NLL

| | $n_{\text{tra}}=25$ | $n_{\text{tra}}=50$ | $n_{\text{tra}}=100$ | $n_{\text{tra}}=200$ | $n_{\text{tra}}=400$ | $n_{\text{tra}}=800$ |
|---|---|---|---|---|---|---|
| BIN | 95, 90 | 93, 91 | 86, 83 | 79, 82 | 67,111 | 49,142 |
| POI | 110, 64 | 111, 76 | 94, 84 | 82, 97 | 64,125 | 41,152 |
| PO-ORD-ACL | 47,152 | 41,150 | 31,147 | 33,145 | 42,151 | 34,149 |
| PO-VS-SL | 60,146 | 65,136 | 57,134 | 44,132 | 49,132 | 55,121 |
| OH-BIN | 134, 46 | 134, 40 | 109, 50 | 73, 76 | 71,110 | 71,102 |
| OH-POI | 147, 23 | 140, 38 | 116, 53 | 92, 78 | 66,113 | 66,118 |
| OH-ORD-ACL | 77, 92 | 74, 93 | 78, 74 | 73, 74 | 72, 96 | 77, 96 |
| OH-VS-SL | 93, 90 | 89, 92 | 91, 73 | 84, 55 | 98, 79 | 118, 63 |
| ORD-ACL | 131, 50 | 142, 33 | 136, 37 | 115, 46 | 134, 38 | 141, 39 |
| VS-SL | 63,125 | 63,130 | 60,117 | 92, 65 | 147, 32 | 168, 25 |
| ACL | 119, 61 | 115, 50 | 126, 36 | 149, 29 | 162, 22 | 156, 29 |
| SL | 35,172 | 34,172 | 49,145 | 73,110 | 109, 72 | 123, 63 |

MAE

| | $n_{\text{tra}}=25$ | $n_{\text{tra}}=50$ | $n_{\text{tra}}=100$ | $n_{\text{tra}}=200$ | $n_{\text{tra}}=400$ | $n_{\text{tra}}=800$ |
|---|---|---|---|---|---|---|
| BIN | 77, 22 | 75, 34 | 63, 44 | 69, 49 | 62, 58 | 62, 71 |
| POI | 92, 23 | 66, 43 | 56, 67 | 38, 90 | 36,104 | 38, 99 |
| PO-ORD-ACL | 55, 48 | 66, 49 | 62, 56 | 54, 60 | 49, 70 | 59, 77 |
| PO-VS-SL | 68, 36 | 82, 37 | 77, 48 | 66, 47 | 56, 55 | 65, 64 |
| OH-BIN | 74, 23 | 81, 28 | 72, 43 | 73, 57 | 65, 71 | 52, 94 |
| OH-POI | 103, 5 | 95, 24 | 95, 53 | 69, 83 | 54, 95 | 49,110 |
| OH-ORD-ACL | 36, 83 | 31, 97 | 33,102 | 21,110 | 37, 95 | 49, 98 |
| OH-VS-SL | 34, 92 | 33,109 | 38,103 | 35,102 | 54, 93 | 51,102 |
| ORD-ACL | 87, 24 | 107, 26 | 128, 23 | 134, 13 | 130, 21 | 146, 32 |
| VS-SL | 49, 66 | 61, 44 | 91, 22 | 120, 20 | 142, 12 | 148, 23 |
| ACL | 37,125 | 44,123 | 57, 94 | 88, 53 | 101, 40 | 125, 44 |
| SL | 0,239 | 7,224 | 13,213 | 22,158 | 39,110 | 71, 93 |
| AD | 100, 26 | 112, 22 | 111, 28 | 96, 43 | 74, 75 | 75, 83 |

MZE

| | $n_{\text{tra}}=25$ | $n_{\text{tra}}=50$ | $n_{\text{tra}}=100$ | $n_{\text{tra}}=200$ | $n_{\text{tra}}=400$ | $n_{\text{tra}}=800$ |
|---|---|---|---|---|---|---|
| BIN | 81, 39 | 67, 53 | 70, 75 | 56,102 | 49,127 | 39,136 |
| POI | 94, 61 | 64,111 | 34,153 | 17,183 | 19,188 | 15,184 |
| PO-ORD-ACL | 58, 50 | 51, 60 | 45, 96 | 37,125 | 38,135 | 42,132 |
| PO-VS-SL | 62, 39 | 74, 50 | 59, 84 | 58,112 | 47,124 | 51,124 |
| OH-BIN | 104, 32 | 116, 22 | 132, 23 | 158, 34 | 159, 42 | 132, 53 |
| OH-POI | 129, 13 | 110, 33 | 121, 45 | 105, 73 | 100, 91 | 85, 98 |
| OH-ORD-ACL | 47, 68 | 50, 82 | 64, 90 | 67, 95 | 81, 95 | 94, 80 |
| OH-VS-SL | 49, 67 | 60, 58 | 86, 65 | 103, 65 | 126, 53 | 131, 50 |
| ORD-ACL | 99, 35 | 124, 27 | 143, 29 | 157, 29 | 135, 35 | 135, 46 |
| VS-SL | 78, 76 | 79, 53 | 107, 42 | 137, 29 | 151, 34 | 145, 29 |
| ACL | 32,159 | 53,140 | 76, 88 | 116, 57 | 130, 54 | 129, 41 |
| SL | 0,240 | 12,221 | 22,197 | 54,149 | 80,106 | 96, 86 |
| AD | 83, 37 | 94, 44 | 96, 68 | 84, 96 | 78,109 | 82,117 |

MSE

| | $n_{\text{tra}}=25$ | $n_{\text{tra}}=50$ | $n_{\text{tra}}=100$ | $n_{\text{tra}}=200$ | $n_{\text{tra}}=400$ | $n_{\text{tra}}=800$ |
|---|---|---|---|---|---|---|
| BIN | 81, 8 | 72, 22 | 66, 23 | 87, 23 | 87, 29 | 89, 37 |
| POI | 94, 14 | 83, 28 | 69, 35 | 59, 35 | 63, 37 | 82, 45 |
| PO-ORD-ACL | 43, 48 | 51, 48 | 52, 44 | 67, 41 | 76, 35 | 92, 32 |
| PO-VS-SL | 52, 53 | 58, 42 | 65, 45 | 74, 31 | 81, 31 | 95, 32 |
| OH-BIN | 91, 11 | 84, 17 | 69, 25 | 46, 69 | 32,106 | 34,136 |
| OH-POI | 112, 3 | 104, 11 | 84, 31 | 54, 66 | 49,104 | 37,123 |
| OH-ORD-ACL | 26, 75 | 28, 74 | 32, 74 | 30, 81 | 53, 76 | 67, 75 |
| OH-VS-SL | 25, 92 | 28, 96 | 32, 85 | 39, 92 | 56, 79 | 58, 94 |
| ORD-ACL | 80, 9 | 85, 20 | 113, 16 | 118, 10 | 120, 23 | 126, 31 |
| VS-SL | 49, 54 | 62, 37 | 82, 27 | 95, 21 | 121, 15 | 124, 21 |
| ACL | 33,110 | 35,106 | 42, 76 | 84, 39 | 92, 38 | 99, 47 |
| SL | 0,225 | 2,213 | 3,194 | 26,151 | 39,121 | 55,107 |
| AD | 61, 45 | 79, 57 | 56, 90 | 23,143 | 20,195 | 18,196 |

## 6.3 Experimental Results

### 6.3.1 Outline of Results

Table 3 shows the results of a comparison of all the methods. In summary, the OH-BIN, OH-POI, and ORD-ACL models were especially good when $n_{\text{tra}}$ was small, the ORD-ACL and VS-SL models were especially good when $n_{\text{tra}}$ was large, for every task. These results imply that methods based on likelihood models with a weak (resp. strong) representation ability are more effective when the training sample size is small (resp. large), and that unimodal models yielded better performances than the ACL and SL models, which are more representable but have no unimodality guarantee, under our implementation of the models (neural network architecture) and within the sample sizes that we considered ($n_{\text{tra}} \leq 800$). These results clarified the significance of the proposed more representable unimodal likelihood models.

### 6.3.2 Detail of Results

For a better understanding of the obtained results, we further provide considerations about comparisons within each set of methods that have interesting differences while maintaining commonality. We believe that these considerations will be helpful for future development; see also Table 4.

**POI *v.s.* PO-ORD-ACL, and OH-POI *v.s.* OH-ORD-ACL**  The POI model is a special instance of the PO-ORD-ACL model with fixed mode-wise heteroscedasticity, and has a weaker representation ability in the formal sense (see Theorem 15). This relation might make the POI model work better when $n_{\text{tra}}$ was small and the PO-ORD-ACL model work better when $n_{\text{tra}}$ was large, in every task. The situation for OH-POI *v.s.* OH-ORD-ACL is similar to that for POI *v.s.* PO-ORD-ACL.

**BIN *v.s.* POI *v.s.* PO-ORD-ACL *v.s.* PO-VS-SL (1-FDF)**  The PO-ORD-ACL and PO-VS-SL models can adjust the mode-wise heteroscedasticity, while the BIN and POI models cannot. In this regard, the PO-ORD-ACL and PO-VS-SL models tend more representable in the informal (or formal) sense than the BIN and POI models. For this reason, it can be considered that the BIN and POI models worked better when $n_{\text{tra}}$ was small and the PO-ORD-ACL and PO-VS-SL models worked better when $n_{\text{tra}}$ was large.

Table 4: A cell for an error and training sample size $n_{\text{tra}}$ in a sub-table for a group of specified methods shows the total (over the 21 datasets) number of times that each method wins in the Mann-Whitney U-test with $p$-value 0.05 regarding the error for each pair of that method and other method in that group. In each cell in a sub-table, methods that win the most times are highlighted in the corresponding color.

POI *v.s.* PO-ORD-ACL

|  | $n_{\text{tra}}=25$ | $n_{\text{tra}}=50$ | $n_{\text{tra}}=100$ | $n_{\text{tra}}=200$ | $n_{\text{tra}}=400$ | $n_{\text{tra}}=800$ |
|---|---|---|---|---|---|---|
| NLL | 12,5 | 12,5 | 13,4 | 14,5 | 11,8 | 7,7 |
| MZE | 6,5 | 5,7 | 3,11 | 2,10 | 3,11 | 1,11 |
| MAE | 5,1 | 4,2 | 1,2 | 1,4 | 2,6 | 2,5 |
| MSE | 7,1 | 6,1 | 3,2 | 1,2 | 2,2 | 1,4 |

PO-ORD-ACL *v.s.* OH-ORD-ACL *v.s.* ORD-ACL

|  | $n_{\text{tra}}=25$ | $n_{\text{tra}}=50$ | $n_{\text{tra}}=100$ | $n_{\text{tra}}=200$ | $n_{\text{tra}}=400$ | $n_{\text{tra}}=800$ |
|---|---|---|---|---|---|---|
| NLL | 7,15,28 | 6,13,28 | 1,14,26 | 3,16,22 | 4,14,30 | 1,14,33 |
| MZE | 6,2,17 | 6,2,22 | 3,7,28 | 2,10,34 | 3,13,27 | 3,16,27 |
| MAE | 7,2,16 | 9,1,22 | 8,2,26 | 9,1,27 | 7,5,22 | 8,7,26 |
| MSE | 2,0,13 | 5,0,15 | 3,1,23 | 9,1,24 | 8,4,18 | 12,4,19 |

BIN *v.s.* POI *v.s.* PO-ORD-ACL *v.s.* PO-VS-SL

|  | $n_{\text{tra}}=25$ | $n_{\text{tra}}=50$ | $n_{\text{tra}}=100$ | $n_{\text{tra}}=200$ | $n_{\text{tra}}=400$ | $n_{\text{tra}}=800$ |
|---|---|---|---|---|---|---|
| NLL | 37,36,17,20 | 37,36,14,23 | 33,34,13,20 | 32,33,13,20 | 29,30,16,20 | 23,24,16,25 |
| MZE | 16,20,6,10 | 16,14,9,16 | 23,7,15,20 | 25,6,13,23 | 24,7,18,26 | 22,6,19,26 |
| MAE | 11,13,2,5 | 7,8,4,7 | 6,3,5,11 | 12,2,5,12 | 13,4,9,11 | 15,5,9,10 |
| MSE | 15,15,2,4 | 8,13,2,5 | 8,8,4,6 | 10,2,4,5 | 5,4,5,5 | 5,4,9,9 |

PO-VS-SL *v.s.* OH-VS-SL *v.s.* VS-SL

|  | $n_{\text{tra}}=25$ | $n_{\text{tra}}=50$ | $n_{\text{tra}}=100$ | $n_{\text{tra}}=200$ | $n_{\text{tra}}=400$ | $n_{\text{tra}}=800$ |
|---|---|---|---|---|---|---|
| NLL | 11,22,17 | 12,24,19 | 11,26,16 | 3,20,21 | 5,15,29 | 4,19,29 |
| MZE | 9,5,10 | 8,5,10 | 4,9,17 | 2,14,25 | 0,18,28 | 1,21,24 |
| MAE | 19,1,9 | 21,0,11 | 14,1,17 | 11,3,22 | 7,5,26 | 10,5,25 |
| MSE | 13,1,10 | 13,0,12 | 11,3,13 | 13,3,15 | 8,3,17 | 12,5,16 |

BIN *v.s.* OH-BIN

|  | $n_{\text{tra}}=25$ | $n_{\text{tra}}=50$ | $n_{\text{tra}}=100$ | $n_{\text{tra}}=200$ | $n_{\text{tra}}=400$ | $n_{\text{tra}}=800$ |
|---|---|---|---|---|---|---|
| NLL | 3,13 | 4,14 | 4,11 | 4,7 | 7,9 | 6,11 |
| MZE | 0,6 | 0,6 | 0,11 | 1,15 | 1,17 | 0,16 |
| MAE | 1,1 | 2,5 | 3,4 | 5,5 | 6,6 | 7,3 |
| MSE | 0,1 | 1,2 | 2,3 | 8,2 | 14,1 | 15,0 |

ORD-ACL *v.s.* ACL

|  | $n_{\text{tra}}=25$ | $n_{\text{tra}}=50$ | $n_{\text{tra}}=100$ | $n_{\text{tra}}=200$ | $n_{\text{tra}}=400$ | $n_{\text{tra}}=800$ |
|---|---|---|---|---|---|---|
| NLL | 9,5 | 8,3 | 8,5 | 5,9 | 2,7 | 3,7 |
| MZE | 16,0 | 17,0 | 11,0 | 8,2 | 5,3 | 5,6 |
| MAE | 15,0 | 14,0 | 12,1 | 8,1 | 8,1 | 6,2 |
| MSE | 14,0 | 14,0 | 10,0 | 7,1 | 8,2 | 5,2 |

POI *v.s.* OH-POI

|  | $n_{\text{tra}}=25$ | $n_{\text{tra}}=50$ | $n_{\text{tra}}=100$ | $n_{\text{tra}}=200$ | $n_{\text{tra}}=400$ | $n_{\text{tra}}=800$ |
|---|---|---|---|---|---|---|
| NLL | 2,7 | 4,10 | 5,10 | 5,11 | 6,11 | 4,14 |
| MZE | 1,8 | 0,12 | 0,16 | 0,18 | 1,17 | 1,18 |
| MAE | 1,6 | 0,7 | 4,9 | 5,8 | 5,8 | 8,6 |
| MSE | 0,3 | 2,6 | 4,6 | 6,3 | 9,3 | 13,1 |

VS-SL *v.s.* SL

|  | $n_{\text{tra}}=25$ | $n_{\text{tra}}=50$ | $n_{\text{tra}}=100$ | $n_{\text{tra}}=200$ | $n_{\text{tra}}=400$ | $n_{\text{tra}}=800$ |
|---|---|---|---|---|---|---|
| NLL | 14,2 | 15,3 | 13,5 | 13,5 | 12,3 | 12,4 |
| MZE | 21,0 | 19,0 | 18,0 | 17,2 | 14,2 | 10,2 |
| MAE | 21,0 | 20,0 | 19,0 | 17,0 | 15,1 | 12,2 |
| MSE | 20,0 | 20,0 | 19,0 | 15,0 | 15,1 | 13,1 |

OH-POI *v.s.* OH-ORD-ACL

|  | $n_{\text{tra}}=25$ | $n_{\text{tra}}=50$ | $n_{\text{tra}}=100$ | $n_{\text{tra}}=200$ | $n_{\text{tra}}=400$ | $n_{\text{tra}}=800$ |
|---|---|---|---|---|---|---|
| NLL | 13,1 | 12,2 | 10,4 | 8,8 | 6,11 | 7,11 |
| MZE | 10,1 | 9,3 | 10,4 | 8,4 | 8,6 | 5,10 |
| MAE | 8,0 | 10,0 | 11,1 | 8,2 | 5,5 | 6,7 |
| MSE | 13,0 | 9,0 | 9,0 | 5,3 | 4,5 | 1,10 |

ORD-ACL *v.s.* VS-SL *v.s.* ACL *v.s.* SL

|  | $n_{\text{tra}}=25$ | $n_{\text{tra}}=50$ | $n_{\text{tra}}=100$ | $n_{\text{tra}}=200$ | $n_{\text{tra}}=400$ | $n_{\text{tra}}=800$ |
|---|---|---|---|---|---|---|
| NLL | 44,16,40,4 | 46,16,36,3 | 41,14,38,8 | 24,16,36,9 | 14,26,30,8 | 14,31,26,10 |
| MZE | 43,37,21,0 | 44,33,20,0 | 40,32,18,0 | 30,29,18,4 | 22,27,20,4 | 21,22,21,5 |
| MAE | 42,29,20,0 | 42,29,19,0 | 37,29,19,0 | 29,28,19,0 | 26,26,15,2 | 25,22,18,3 |
| MSE | 40,28,19,0 | 36,30,19,0 | 37,28,14,0 | 29,21,18,0 | 27,25,14,2 | 24,24,14,3 |

OH-BIN *v.s.* OH-POI *v.s.* OH-ORD-ACL *v.s.* OH-VS-SL

|  | $n_{\text{tra}}=25$ | $n_{\text{tra}}=50$ | $n_{\text{tra}}=100$ | $n_{\text{tra}}=200$ | $n_{\text{tra}}=400$ | $n_{\text{tra}}=800$ |
|---|---|---|---|---|---|---|
| NLL | 28,37,8,10 | 32,35,9,13 | 23,28,13,13 | 14,21,17,19 | 15,15,24,30 | 16,14,23,36 |
| MZE | 22,30,4,4 | 23,24,5,9 | 29,27,6,16 | 36,18,6,16 | 34,12,12,24 | 26,12,15,25 |
| MAE | 19,22,3,2 | 22,26,3,2 | 19,26,4,4 | 25,20,4,6 | 19,12,12,11 | 17,14,17,13 |
| MSE | 23,29,2,1 | 20,23,3,2 | 17,22,2,2 | 13,14,7,7 | 7,12,13,11 | 9,10,24,18 |

**BIN *v.s.* OH-BIN, and POI *v.s.* OH-POI**   The OH-BIN and OH-POI models are respectively OH-generalizations of the BIN and POI models, and have a stronger representation ability in the formal sense. Thus, OH models were better regarding the NLL that was directly optimized. However, OH models were bad for some tasks (especially, task-A and -S) when $n_{\text{tra}}$ was large, which is a counter-intuitive result and requires further analysis but may be because the difference of the restricted structures of the two likelihood models makes a difference in the compatibility between the model and task.

**OH-BIN *v.s.* OH-POI *v.s.* OH-ORD-ACL *v.s.* OH-VS-SL (2-FDF)**   The situation is similar to that for the 1-FDF models, BIN *v.s.* POI *v.s.* PO-ORD-ACL *v.s.* PO-VS-SL.

**PO-ORD-ACL *v.s.* OH-ORD-ACL *v.s.* ORD-ACL, and PO-VS-SL *v.s.* OH-VS-SL *v.s.* VS-SL**   The VS-SL, OH-VS-SL, and PO-VS-SL models (and the ORD-ACL, OH-ORD-ACL, and PO-ORD-ACL models) are more representable in the formal sense in that order. Presumably for this reason, the PO- and OH-VS-SL models worked better with small $n_{\text{tra}}$, and the VS-SL model worked better with large $n_{\text{tra}}$.

**ORD-ACL *v.s.* ACL**   The ACL model can represent any data (see Theorem 2), and the ORD-ACL model can represent unimodal data with the DRs' unimodality constraint (see Theorem 4). The unimodality of the ORD-ACL model might improve the generalization performance when $n_{\text{tra}}$ was small, but the ORD-ACL model was not good when $n_{\text{tra}}$ was large perhaps because its DRs' unimodality constraint restricted its representation ability too much for our tried data; see again DRs' MU in Tables 1 and 2.

**VS-SL *v.s.* SL**    The SL model can represent any data (see Theorem 5), and the VS-SL model can represent any unimodal data (see Theorem 7). Unlike the comparison between ACL and ORD-ACL models, the VS-SL model gave better results than the SL model, probably thanks to the validity of the unimodal hypothesis.

**ORD-ACL *v.s.* VS-SL *v.s.* ACL *v.s.* SL (Full-FDF)**    The ORD-ACL model was the best in many cases, and the VS-SL model was better when $n_{\mathrm{tra}}$ was large. This is thought due to the unimodality guarantee of these models and the DRs' unimodality constraint of the ORD-ACL model.

For further details of the experimental results, refer to Appendix C.

## 7  Conclusion

In this paper, we pointed out that previous unimodal BIN and POI models have a weak representation ability from the perspective of the DRs' unimodality constraint, scale, skewness, and FDF, and then developed more representable unimodal ORD-ACL and VS-SL models as well as their PO-constrained version and OH-extension. In our experiments, 1-FDF or 2-FDF OH models worked better when the training sample size was small, and full-FDF ORD-ACL and VS-SL models worked better than low-FDF unimodal models when the training sample size was large. Also, full-FDF ORD-ACL and VS-SL models were better than full-FDF models with no unimodality guarantee, due to their unimodality guarantee reasonable for many real-world data that were almost unimodal.

Finally, we have to mention one caution, for future research. We did not encounter non-unimodal ordinal data in this study, but not all ordinal data would be almost-unimodal. If you feel that the data you are interested in have a natural ordinal relation and consider to apply unimodal likelihood models, it would be better to first test the unimodality of the data as we did with the MU in our experiments. As is obvious, unimodal likelihood models would not work well if the data do not tend unimodal.

### Acknowledgments

This work was supported by Grant-in-Aid for JSPS Fellows, Number 20J23367.

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

## A  Proof of Theorems in Main Part

**BIN and OH-BIN**  We first prove Theorem 1 about properties of the BIN and OH-BIN models:

*Proof of Theorem 1.* First, we prove Theorem 1 (i): Letting $p = \frac{1}{1+e^{-u}}$, one has that

$$
\begin{aligned}
\frac{P_{\text{bin}}(y+1; u, s)}{P_{\text{bin}}(y; u, s)} &= \frac{e^{\log(P_{\text{b}}(y+1;p))/s}}{\sum_{k=1}^{K} e^{\log(P_{\text{b}}(k;p))/s}} \bigg/ \frac{e^{\log(P_{\text{b}}(y;p))/s}}{\sum_{k=1}^{K} e^{\log(P_{\text{b}}(k;p))/s}} = e^{\{\log(P_{\text{b}}(y+1;p))-\log(P_{\text{b}}(y;p))\}/s} \\
&= e^{\{\log(\frac{(K-1)!}{y!(K-y-1)!} p^y (1-p)^{K-y-1})-\log(\frac{(K-1)!}{(y-1)!(K-y)!} p^{y-1} (1-p)^{K-y})\}/s} = \left( \frac{(K-y)p}{y(1-p)} \right)^{1/s}.
\end{aligned}
\tag{21}
$$

Next, we prove Theorem 1 (ii) and (iii): Let $r(y,p) := \frac{(K-y)p}{y(1-p)}$ so that $\frac{P_{\text{bin}}(y+1;u,s)}{P_{\text{bin}}(y;u,s)} = \{r(y,p)\}^{1/s}$. (6) shows $\max\{1, Kp\} \leq M((P_{\text{b}}(k;p))_{k \in [K]}) \leq \min\{Kp+1, K\}$ with $p = \frac{1}{1+e^{-u}}$. Since $(P_{\text{b}}(k;p))_{k \in [K]} = (P_{\text{bin}}(y;u,1))_{y \in [K]}$ and the scaling factor does not change the mode, one has that $\max\{1, Kp\} \leq m = M((P_{\text{bin}}(y;u,s))_{y \in [K]}) \leq \min\{Kp+1, K\}$. This implies $Kp \leq m \leq Kp+1$ (or equivalently $(m-1)/K \leq p \leq m/K$). Since $r(y,p)$ is decreasing in $y \in [K-1]$ and increasing in $p \in (0,1)$, one has that

$$
\begin{aligned}
r(y,p) &= \frac{(K-y)p}{y(1-p)} \geq \left.\frac{(K-y)p}{y(1-p)}\right|_{y=m-1, p=(m-1)/K} = 1 \text{ for } y = 1, \ldots, m-1, \\
r(y,p) &= \frac{(K-y)p}{y(1-p)} \leq \left.\frac{(K-y)p}{y(1-p)}\right|_{y=m, p=m/K} = 1 \text{ for } y = m, \ldots, K-1,
\end{aligned}
\tag{22}
$$

i.e., Theorem 1 (ii). Also, these results and the fact that $r(y,p)$ is decreasing in $y \in [K-1]$ prove that

$$
r(1,p)^{-1} \leq r(2,p)^{-1} \leq \cdots \leq r(m-1,p)^{-1} \leq 1, \text{ and } 1 \geq r(m,p) \geq r(m+1,p) \geq \cdots \geq r(K-1,p), \tag{23}
$$

which imply Theorem 1 (iii). □

**ACL and ORD-ACL and their PO and OH versions** We next provide proofs of Theorems 2, 3, and 4 about properties of the ACL and ORD-ACL models and their PO and OH versions:

*Proof of Theorem 2.* The construction of the ACL model trivially shows $\{\boldsymbol{p} : \mathbb{R}^d \to \bar{\Delta}_{K-1}\} \supseteq \{(P_{\text{acl}}(y; \boldsymbol{g}(\cdot)))_{y \in [K]} \mid \boldsymbol{g} : \mathbb{R}^d \to \mathbb{R}^{K-1}\}$, namely, that $(P_{\text{acl}}(y; \boldsymbol{g}(\boldsymbol{x})))_{y \in [K]} \in \bar{\Delta}_{K-1}$ for any $\boldsymbol{g} : \mathbb{R}^d \to \mathbb{R}^{K-1}$ and $\boldsymbol{x} \in \mathbb{R}^d$. Thus, we here show $\{\boldsymbol{p} : \mathbb{R}^d \to \bar{\Delta}_{K-1}\} \subseteq \{(P_{\text{acl}}(y; \boldsymbol{g}(\cdot)))_{y \in [K]} \mid \boldsymbol{g} : \mathbb{R}^d \to \mathbb{R}^{K-1}\}$, namely, that there exists $\boldsymbol{g} : \mathbb{R}^d \to \mathbb{R}^{K-1}$ such that $\boldsymbol{p}(\cdot) = (P_{\text{acl}}(y; \boldsymbol{g}(\cdot)))_{y \in [K]}$ for any $\boldsymbol{p} : \mathbb{R}^d \to \bar{\Delta}_{K-1}$. Considering the proportional expression

$$
\begin{aligned}
p_1(\boldsymbol{x}) : p_2(\boldsymbol{x}) : \cdots : p_K(\boldsymbol{x}) &= P_{\text{acl}}(1; \boldsymbol{g}(\boldsymbol{x})) : P_{\text{acl}}(2; \boldsymbol{g}(\boldsymbol{x})) : \cdots : P_{\text{acl}}(K; \boldsymbol{g}(\boldsymbol{x})) \\
&= e^{-\sum_{l=1}^{1-1} g_l(\boldsymbol{x})} : e^{-\sum_{l=1}^{2-1} g_l(\boldsymbol{x})} : \cdots : e^{-\sum_{l=1}^{K-1} g_l(\boldsymbol{x})},
\end{aligned}
\tag{24}
$$

we can find that the equation $\boldsymbol{p}(\cdot) = (P_{\text{acl}}(y; \boldsymbol{g}(\cdot)))_{y \in [K]}$ is satisfied by using $\boldsymbol{g}$ such that

$$
g_k(\boldsymbol{x}) = \log \frac{p_k(\boldsymbol{x})}{p_{k+1}(\boldsymbol{x})} \in \mathbb{R} \text{ for } k = 1, \ldots, K-1. \tag{25}
$$

This concludes the proof. □

*Proof of Theorem 3.* First, Theorem 3 (i) is proved by

$$
\frac{P_{\text{acl}}(y+1; \boldsymbol{u})}{P_{\text{acl}}(y; \boldsymbol{u})} = \frac{\prod_{l=1}^{y} e^{-u_l}}{\sum_{k=1}^{K} \prod_{l=1}^{k-1} e^{-u_l}} \bigg/ \frac{\prod_{l=1}^{y-1} e^{-u_l}}{\sum_{k=1}^{K} \prod_{l=1}^{k-1} e^{-u_l}} = \frac{\prod_{l=1}^{y} e^{-u_l}}{\prod_{l=1}^{y-1} e^{-u_l}} = e^{-u_y}. \tag{26}
$$

Next, we prove Theorem 3 (ii) and (iii): Under the assumption that $u_0(:= -\infty) \leq \cdots \leq u_{m-1} \leq 0 \leq u_m \leq \cdots \leq u_K(:= \infty)$ for some $m \in [K]$, one has that

$$
\begin{aligned}
\frac{P_{\text{acl}}(1; \boldsymbol{u})}{P_{\text{acl}}(2; \boldsymbol{u})} &= e^{u_1} \leq \frac{P_{\text{acl}}(2; \boldsymbol{u})}{P_{\text{acl}}(3; \boldsymbol{u})} = e^{u_2} \leq \cdots \leq \frac{P_{\text{acl}}(m-1; \boldsymbol{u})}{P_{\text{acl}}(m; \boldsymbol{u})} = e^{u_{m-1}} \leq 1, \\
1 &\geq \frac{P_{\text{acl}}(m+1; \boldsymbol{u})}{P_{\text{acl}}(m; \boldsymbol{u})} = e^{-u_m} \geq \frac{P_{\text{acl}}(m+2; \boldsymbol{u})}{P_{\text{acl}}(m+1; \boldsymbol{u})} = e^{-u_{m+1}} \geq \cdots \geq \frac{P_{\text{acl}}(K; \boldsymbol{u})}{P_{\text{acl}}(K-1; \boldsymbol{u})} = e^{-u_{K-1}}.
\end{aligned}
\tag{27}
$$

These results show Theorem 3 (ii) and (iii). □

*Proof of Theorem 4.* Theorem 3 and the fact that $\acute{\boldsymbol{g}} = \rho[\boldsymbol{g}]$ is ordered ($\acute{g}_1(\boldsymbol{x}) \leq \cdots \leq \acute{g}_{K-1}(\boldsymbol{x})$ for any $\boldsymbol{x} \in \mathbb{R}^d$) show 'left-hand side of (14)' $\subseteq$ 'right-hand side of (14)'. Thus, we here show 'left-hand side of (14)' $\supseteq$

'right-hand side of (14)', namely, that there exists $\boldsymbol{g} : \mathbb{R}^d \to \mathbb{R}^{K-1}$ such that $\boldsymbol{p}(\cdot) = (P_{\mathrm{acl}}(y; \acute{\boldsymbol{g}}(\cdot)))_{y \in [K]}$ with $\acute{\boldsymbol{g}} = \rho[\boldsymbol{g}]$ for any $\boldsymbol{p} \in$ 'right-hand side of (14)'. Considering the proportional expression

$$
\begin{aligned}
p_1(\boldsymbol{x}) : p_2(\boldsymbol{x}) : \cdots : p_K(\boldsymbol{x}) &= P_{\mathrm{acl}}(1; \acute{\boldsymbol{g}}(\boldsymbol{x})) : P_{\mathrm{acl}}(2; \acute{\boldsymbol{g}}(\boldsymbol{x})) : \cdots : P_{\mathrm{acl}}(K; \acute{\boldsymbol{g}}(\boldsymbol{x})) \\
&= e^{-\sum_{l=1}^{1-1} \acute{g}_l(\boldsymbol{x})} : e^{-\sum_{l=1}^{2-1} \acute{g}_l(\boldsymbol{x})} : \cdots : e^{-\sum_{l=1}^{K-1} \acute{g}_l(\boldsymbol{x})},
\end{aligned}
\tag{28}
$$

we can find that the equation $\boldsymbol{p}(\cdot) = (P_{\mathrm{acl}}(y; \acute{\boldsymbol{g}}(\cdot)))_{y \in [K]}$ is satisfied by using $\boldsymbol{g}$ such that

$$
g_1(\boldsymbol{x}) = \acute{g}_1(\boldsymbol{x}) = \log \frac{p_1(\boldsymbol{x})}{p_2(\boldsymbol{x})} \in \mathbb{R},
$$

$$
\rho(g_k(\boldsymbol{x})) = \acute{g}_k(\boldsymbol{x}) - \acute{g}_{k-1}(\boldsymbol{x}) = \log \frac{p_k(\boldsymbol{x})}{p_{k+1}(\boldsymbol{x})} - \log \frac{p_{k-1}(\boldsymbol{x})}{p_k(\boldsymbol{x})} \text{ for } k = 2, \ldots, K-1.
\tag{29}
$$

The question here is whether there exists $g_k : \mathbb{R}^d \to \mathbb{R}$ satisfying the latter half of (29). The DRs' unimodality shows that $\frac{p_{k-1}(\boldsymbol{x})}{p_k(\boldsymbol{x})} \le \frac{p_k(\boldsymbol{x})}{p_{k+1}(\boldsymbol{x})}$. Thus, $\log \frac{p_k(\boldsymbol{x})}{p_{k+1}(\boldsymbol{x})} - \log \frac{p_{k-1}(\boldsymbol{x})}{p_k(\boldsymbol{x})} \in [0, \infty)$, and there exists $g_k : \mathbb{R}^d \to \mathbb{R}$ satisfying the latter half of (29) for a non-negative function $\rho$ satisfying (13). This concludes the proof. □

**SL and VS-SL and their PO and OH versions** We finally provide proofs of Theorems 5, 6, and 7 about properties of the SL, VS-SL, and PO- and OH-VS-SL models:

*Proof of Theorem 5.* The equation $(P_{\mathrm{sl}}(y; \boldsymbol{g}(\boldsymbol{x})))_{y \in [K]} = (P_{\mathrm{sl}}(y; \boldsymbol{g}(\boldsymbol{x}) + c \cdot \mathbf{1}))_{y \in [K]}$, $c \in \mathbb{R}$ shows the former equation of (17). We thus prove the latter equation of (17) below. The construction of the SL model trivially shows $\{\boldsymbol{p} : \mathbb{R}^d \to \bar{\Delta}_{K-1}\} \supseteq \{(P_{\mathrm{sl}}(y; \boldsymbol{g}(\cdot)))_{y \in [K]} \mid \boldsymbol{g} : \mathbb{R}^d \to \mathbb{R}^K\}$, namely, that $(P_{\mathrm{sl}}(y; \boldsymbol{g}(\boldsymbol{x})))_{y \in [K]} \in \bar{\Delta}_{K-1}$ for any $\boldsymbol{g} : \mathbb{R}^d \to \mathbb{R}^K$ and $\boldsymbol{x} \in \mathbb{R}^d$. Thus, we here show $\{\boldsymbol{p} : \mathbb{R}^d \to \bar{\Delta}_{K-1}\} \subseteq \{(P_{\mathrm{sl}}(y; \boldsymbol{g}(\cdot)))_{y \in [K]} \mid \boldsymbol{g} : \mathbb{R}^d \to \mathbb{R}^K\}$, namely, that there exists $\boldsymbol{g} : \mathbb{R}^d \to \mathbb{R}^K$ such that $\boldsymbol{p}(\cdot) = (P_{\mathrm{sl}}(y; \boldsymbol{g}(\cdot)))_{y \in [K]}$ for any $\boldsymbol{p} : \mathbb{R}^d \to \bar{\Delta}_{K-1}$. Considering the proportional expression

$$
p_1(\boldsymbol{x}) : p_2(\boldsymbol{x}) : \cdots : p_K(\boldsymbol{x}) = P_{\mathrm{sl}}(1; \boldsymbol{g}(\boldsymbol{x})) : P_{\mathrm{sl}}(2; \boldsymbol{g}(\boldsymbol{x})) : \cdots : P_{\mathrm{sl}}(K; \boldsymbol{g}(\boldsymbol{x})) = e^{-g_1(\boldsymbol{x})} : e^{-g_2(\boldsymbol{x})} : \cdots : e^{-g_K(\boldsymbol{x})}, \tag{30}
$$

we can find that the equation $\boldsymbol{p}(\cdot) = (P_{\mathrm{sl}}(y; \boldsymbol{g}(\cdot)))_{y \in [K]}$ is satisfied by using $\boldsymbol{g}$ such that

$$
g_k(\boldsymbol{x}) = -\log p_k(\boldsymbol{x}) \in \mathbb{R} \text{ for } k = 1, \ldots, K-1. \tag{31}
$$

This concludes the proof. □

The first equation of (17) suggests that the presence or absence of the constraint $g_1(\cdot) = 0$ in the SL model is not related to the representation ability of the SL model. So we adopt the formulation of the SL model without the constraint $g_1(\cdot) = 0$ in the following.

*Proof of Theorem 6.* Theorem 6 would be trivial. We omit the proof. □

*Proof of Theorem 7.* Theorem 6 and the fact that $\check{\boldsymbol{g}} = \tau(\rho[\boldsymbol{g}])$ is V-shaped (there exists $m(\boldsymbol{x}) \in [K]$ such that $\check{g}_1(\boldsymbol{x}) \ge \cdots \ge \check{g}_{m(\boldsymbol{x})}(\boldsymbol{x})$ and $\check{g}_{m(\boldsymbol{x})}(\boldsymbol{x}) \le \cdots \le \check{g}_K(\boldsymbol{x})$ for any $\boldsymbol{x} \in \mathbb{R}^d$) show 'left-hand side of (19)' $\subseteq$ 'right-hand side of (19)'. Thus, we here show 'left-hand side of (19)' $\supseteq$ 'right-hand side of (19)', namely, that there exists $\boldsymbol{g} : \mathbb{R}^d \to \mathbb{R}^K$ such that $\boldsymbol{p}(\cdot) = (P_{\mathrm{acl}}(y; \acute{\boldsymbol{g}}(\cdot)))_{y \in [K]}$ for any $\boldsymbol{p} \in$ 'right-hand side of (19)'. Considering the proportional expression

$$
p_1(\boldsymbol{x}) : p_2(\boldsymbol{x}) : \cdots : p_K(\boldsymbol{x}) = P_{\mathrm{sl}}(1; \check{\boldsymbol{g}}(\boldsymbol{x})) : P_{\mathrm{sl}}(2; \check{\boldsymbol{g}}(\boldsymbol{x})) : \cdots : P_{\mathrm{sl}}(K; \check{\boldsymbol{g}}(\boldsymbol{x})) = e^{-\check{g}_1(\boldsymbol{x})} : e^{-\check{g}_2(\boldsymbol{x})} : \cdots : e^{-\check{g}_K(\boldsymbol{x})}, \tag{32}
$$

we can find that the equation $\boldsymbol{p}(\cdot) = (P_{\mathrm{sl}}(y; \check{\boldsymbol{g}}(\cdot)))_{y \in [K]}$ is satisfied by using $\boldsymbol{g}$ such that

$$
\tau(\acute{g}_{k+1}(\boldsymbol{x})) - \tau(\acute{g}_k(\boldsymbol{x})) = \begin{cases} \tau(g_1(\boldsymbol{x}) + \rho(g_2(\boldsymbol{x}))) - \tau(g_1(\boldsymbol{x})) = \log \frac{p_1(\boldsymbol{x})}{p_2(\boldsymbol{x})}, & k = 1, \\ \tau(\acute{g}_k(\boldsymbol{x}) + \rho(g_{k+1}(\boldsymbol{x}))) - \tau(\acute{g}_k(\boldsymbol{x})) = \log \frac{p_k(\boldsymbol{x})}{p_{k+1}(\boldsymbol{x})}, & k = 2, \ldots, K-1. \end{cases} \tag{33}
$$

For simplicity, let $\acute{g}_{M(\boldsymbol{p}(\boldsymbol{x}))}(\boldsymbol{x}) = 0$ (which is achievable for the VS-SL model). Since $\rho$ is non-negative and $\tau(u)$ is continuous and non-increasing in $u < 0$, one has that $\{\tau(c) - \tau(c - \rho(u)) \mid u \in \mathbb{R}\} = \{\tau(c) - \tau(c - v) \mid v \geq 0\} = (-\infty, 0] \ni \log \frac{p_{M(\boldsymbol{p}(\boldsymbol{x}))-1}(\boldsymbol{x})}{p_{M(\boldsymbol{p}(\boldsymbol{x}))}(\boldsymbol{x})} (\leq 0)$ (here $c = \acute{g}_{M(\boldsymbol{p}(\boldsymbol{x}))}(\boldsymbol{x})$), which implies that there exists $g_{M(\boldsymbol{p}(\boldsymbol{x}))}(\boldsymbol{x})$ satisfying (33). By repeating the same argument, one can also prove the existence of $g_1(\boldsymbol{x}), \ldots, g_{M(\boldsymbol{p}(\boldsymbol{x}))-1}(\boldsymbol{x})$ satisfying (33). Since $\rho$ is non-negative and $\tau(u)$ is continuous and non-decreasing in $u > 0$, one has that $\{\tau(c + \rho(u)) - \tau(c) \mid u \in \mathbb{R}\} = \{\tau(c + v) - \tau(c) \mid v \geq 0\} = [0, \infty) \ni \log \frac{p_{M(\boldsymbol{p}(\boldsymbol{x}))}(\boldsymbol{x})}{p_{M(\boldsymbol{p}(\boldsymbol{x}))+1}(\boldsymbol{x})} (\geq 0)$ (here $c = \acute{g}_{M(\boldsymbol{p}(\boldsymbol{x}))}(\boldsymbol{x})$), which implies that there exists $g_{M(\boldsymbol{p}(\boldsymbol{x}))+1}(\boldsymbol{x})$ satisfying (33). By repeating the same argument, one can also prove the existence of $g_{M(\boldsymbol{p}(\boldsymbol{x}))+2}(\boldsymbol{x}), \ldots, g_K(\boldsymbol{x})$ satisfying (33). This concludes the proof. □

# B   Theorems on Relationship among Likelihood Models and their Proof

We here provide theorems on relationships among likelihood models, which are shown in Figure 1. Many of the theorems are direct corollaries of Theorems 1–7.

**ACL = SL**   Theorems 2 and 5 show the following theorem:

**Theorem 8.** *The ACL and SL models have an equivalent representation ability in the formal sense: it holds that*

$$\{(P_{\mathrm{acl}}(y; \boldsymbol{g}(\cdot)))_{y \in [K]} \mid \boldsymbol{g} : \mathbb{R}^d \to \mathbb{R}^{K-1}\} = \{(P_{\mathrm{sl}}(y; \boldsymbol{g}(\cdot)))_{y \in [K]} \mid \boldsymbol{g} : \mathbb{R}^d \to \mathbb{R}^K\}. \tag{34}$$

*Proof of Theorem 8.* This theorem is a direct corollary of Theorems 2 and 5. □

**VS-SL$\subseteq$ SL and ACL**   The following theorem would be trivial from Theorems 2, 5, and 7:

**Theorem 9.** *The VS-SL model has a weak representation ability in the formal sense than the ACL and SL models: for any functions $\rho$ satisfying (13) and $\tau$ satisfying (18), it holds that*

$$\{(P_{\mathrm{sl}}(y; \tau(\rho[\boldsymbol{g}(\cdot)])))_{y \in [K]} \mid \boldsymbol{g} : \mathbb{R}^d \to \mathbb{R}^K\} \subseteq \{(P_{\mathrm{acl}}(y; \boldsymbol{g}(\cdot)))_{y \in [K]} \mid \boldsymbol{g} : \mathbb{R}^d \to \mathbb{R}^{K-1}\}, \tag{35}$$

$$\{(P_{\mathrm{sl}}(y; \tau(\rho[\boldsymbol{g}(\cdot)])))_{y \in [K]} \mid \boldsymbol{g} : \mathbb{R}^d \to \mathbb{R}^K\} \subseteq \{(P_{\mathrm{sl}}(y; \boldsymbol{g}(\cdot)))_{y \in [K]} \mid \boldsymbol{g} : \mathbb{R}^d \to \mathbb{R}^K\}. \tag{36}$$

*Proof of Theorem 9.* The trivial result, $(P_{\mathrm{sl}}(y; \tau(\rho[\boldsymbol{g}(\boldsymbol{x})])))_{y \in [K]} \in \bar{\Delta}_{K-1}$, and Theorems 2, 5, and 7 prove this theorem. □

**ORD-ACL $\subseteq$ VS-SL**   Theorems 4 and 7 show the following theorem:

**Theorem 10.** *The ORD-ACL model has a weak representation ability in the formal sense than the VS-SL model: for any functions $\rho$ satisfying (13) and $\tau$ satisfying (18), it holds that*

$$\{(P_{\mathrm{acl}}(y; \rho[\boldsymbol{g}(\cdot)]))_{y \in [K]} \mid \boldsymbol{g} : \mathbb{R}^d \to \mathbb{R}^{K-1}\} \subseteq \{(P_{\mathrm{sl}}(y; \tau(\rho[\boldsymbol{g}(\cdot)])))_{y \in [K]} \mid \boldsymbol{g} : \mathbb{R}^d \to \mathbb{R}^K\}. \tag{37}$$

*Proof of Theorem 10.* Since $\boldsymbol{p} \in \bar{\Delta}_{K-1}$ satisfies

$$\left(\frac{p_1}{p_2}, \ldots, \frac{p_{M(\boldsymbol{p})-1}}{p_{M(\boldsymbol{p})}}, \frac{p_{M(\boldsymbol{p})+1}}{p_{M(\boldsymbol{p})}}, \ldots, \frac{p_K}{p_{K-1}}\right) \text{ is unimodal} \Rightarrow \boldsymbol{p} \text{ is unimodal}, \tag{38}$$

Theorems 4 and 7 show this theorem. □

**OH-BIN $\subseteq$ ORD-ACL**   Theorems 1 and 4 show the following theorem:

**Theorem 11.** *The OH-BIN model has a weak representation ability in the formal sense than the ORD-ACL model: for any function $\rho$ satisfying (13), it holds that*

$$\{(P_{\mathrm{bin}}(y; g(\cdot), s(\cdot)))_{y \in [K]} \mid g : \mathbb{R}^d \to \mathbb{R}, s : \mathbb{R}^d \to (0, \infty)\} \subseteq \{(P_{\mathrm{acl}}(y; \rho[\boldsymbol{g}(\cdot)]))_{y \in [K]} \mid \boldsymbol{g} : \mathbb{R}^d \to \mathbb{R}^{K-1}\}. \tag{39}$$

*Proof of Theorem 11.* Theorem 1 (iii) shows that an instance of the OH-BIN model has the DRs' unimodality constraint, and Theorem 4 shows that the ORD-ACL model can represent arbitrary CPD with the DRs' unimodality constraint. According to these results, this theorem is proved. □

**OH-ORD-ACL ⊆ ORD-ACL**    Theorems 3 and 4 show the following theorem:

**Theorem 12.** *The OH-ORD-ACL model has a weak representation ability in the formal sense than the ORD-ACL model: for any function $\rho$ satisfying* (13), *it holds that*

$$\{(P_{\mathrm{acl}}(y; \{\rho[\boldsymbol{b}] - g(\cdot) \cdot \mathbf{1}\}/s(\cdot)))_{y \in [K]} \mid \boldsymbol{b} \in \mathbb{R}^{K-1}, g : \mathbb{R}^d \to \mathbb{R}, s : \mathbb{R}^d \to (0, \infty)\}$$
$$\subseteq \{(P_{\mathrm{acl}}(y; \rho[\boldsymbol{g}(\cdot)]))_{y \in [K]} \mid \boldsymbol{g} : \mathbb{R}^d \to \mathbb{R}^{K-1}\}. \tag{40}$$

*Proof of Theorem 12.* Theorem 3 (iii) shows that an instance of the OH-ORD-ACL model has the DRs' unimodality constraint since $\{\rho[\boldsymbol{b}] - g(\cdot) \cdot \mathbf{1}\}/s(\cdot)$ is ordered, and Theorem 4 shows that the ORD-ACL model can represent arbitrary CPD with the DRs' unimodality constraint. These results prove this theorem.    □

**OH-VS-SL ⊆ VS-SL**    Theorems 6 and 7 show the following theorem:

**Theorem 13.** *The OH-VS-SL model has a weak representation ability in the formal sense than the VS-SL model: for any functions $\rho$ satisfying* (13) *and $\tau$ satisfying* (18), *it holds that*

$$\{(P_{\mathrm{sl}}(y; \tau(\rho[\boldsymbol{b}] - g(\cdot) \cdot \mathbf{1})/s(\cdot)))_{y \in [K]} \mid \boldsymbol{b} \in \mathbb{R}^K, g : \mathbb{R}^d \to \mathbb{R}, s : \mathbb{R}^d \to (0, \infty)\}$$
$$\subseteq \{(P_{\mathrm{sl}}(y; \tau(\rho[\boldsymbol{g}(\cdot)])))_{y \in [K]} \mid \boldsymbol{g} : \mathbb{R}^d \to \mathbb{R}^K\}. \tag{41}$$

*Proof of Theorem 13.* Theorem 6 shows that an instance of the OH-VS-SL model is unimodal since $\tau(\rho[\boldsymbol{b}] - g(\cdot) \cdot \mathbf{1})/s(\cdot)$ is V-shaped, and Theorem 7 shows that the VS-SL model can represent arbitrary unimodal CPD. According to these results, this theorem is proved.    □

**BIN ⊆ OH-BIN, etc.**    The following theorem would be trivial:

**Theorem 14.** *For any functions $\rho$ satisfying* (13) *and $\tau$ satisfying* (18), *it holds that*

- *The BIN model has a weak representation ability in the formal sense than the OH-BIN model:*

$$\{(P_{\mathrm{bin}}(y; g(\cdot), s))_{y \in [K]} \mid g : \mathbb{R}^d \to \mathbb{R}, s \in (0, \infty)\}$$
$$\subseteq \{(P_{\mathrm{bin}}(y; g(\cdot), s(\cdot)))_{y \in [K]} \mid g : \mathbb{R}^d \to \mathbb{R}, s : \mathbb{R}^d \to (0, \infty)\}. \tag{42}$$

- *The POI model has a weak representation ability in the formal sense than the OH-POI model:*

$$\{(P_{\mathrm{poi}}(y; g(\cdot), s))_{y \in [K]} \mid g : \mathbb{R}^d \to \mathbb{R}, s \in (0, \infty)\}$$
$$\subseteq \{(P_{\mathrm{poi}}(y; g(\cdot), s(\cdot)))_{y \in [K]} \mid g : \mathbb{R}^d \to \mathbb{R}, s : \mathbb{R}^d \to (0, \infty)\}. \tag{43}$$

- *The PO-ORD-ACL model has a weak representation ability in the formal sense than the OH-ORD-ACL model:*

$$\{(P_{\mathrm{acl}}(y; \rho[\boldsymbol{b}] - g(\cdot) \cdot \mathbf{1}))_{y \in [K]} \mid \boldsymbol{b} \in \mathbb{R}^{K-1}, g : \mathbb{R}^d \to \mathbb{R}\}$$
$$\subseteq \{(P_{\mathrm{acl}}(y; \{\rho[\boldsymbol{b}] - g(\cdot) \cdot \mathbf{1}\}/s(\cdot)))_{y \in [K]} \mid \boldsymbol{b} \in \mathbb{R}^{K-1}, g : \mathbb{R}^d \to \mathbb{R}, s : \mathbb{R}^d \to (0, \infty)\}. \tag{44}$$

- *The PO-VS-SL model has a weak representation ability in the formal sense than the OH-VS-SL model:*

$$\{(P_{\mathrm{sl}}(y; \tau(\rho[\boldsymbol{b}] - g(\cdot) \cdot \mathbf{1})))_{y \in [K]} \mid \boldsymbol{b} \in \mathbb{R}^K, g : \mathbb{R}^d \to \mathbb{R}\}$$
$$\subseteq \{(P_{\mathrm{sl}}(y; \tau(\rho[\boldsymbol{b}] - g(\cdot) \cdot \mathbf{1})/s(\cdot)))_{y \in [K]} \mid \boldsymbol{b} \in \mathbb{R}^K, g : \mathbb{R}^d \to \mathbb{R}, s : \mathbb{R}^d \to (0, \infty)\}. \tag{45}$$

*Proof of Theorem 14.* The BIN, POI, PO-ORD-ACL, and PO-VS-SL models are respectively a special instance of the OH-BIN, OH-POI, OH-ORD-ACL, and OH-VS-SL models with fixing $s(\boldsymbol{x})$ to a $\boldsymbol{x}$-independent constant ($s$ or 1).    □

**POI ⊆ PO-ORD-ACL, etc.** The following theorem formalizes (20), and show the relationship between POI and PO-ORD-ACL models and its OH version.

**Theorem 15.** (20) *holds. Also, for any function $\rho$ satisfying* (13), *it holds that*

- *The POI model has a weak representation ability in the formal sense than the PO-ORD-ACL model:*

$$
\{(P_{\text{poi}}(y; g(\cdot), s))_{y \in [K]} \mid g : \mathbb{R}^d \to \mathbb{R}, s \in (0, \infty)\}
$$
$$
\subseteq \{(P_{\text{acl}}(y; \rho[\boldsymbol{b}] - g(\cdot) \cdot \mathbf{1}))_{y \in [K]} \mid \boldsymbol{b} \in \mathbb{R}^{K-1}, g : \mathbb{R}^d \to \mathbb{R}\}. \tag{46}
$$

- *The OH-POI model has a weak representation ability in the formal sense than the OH-ORD-ACL model:*

$$
\{(P_{\text{poi}}(y; g(\cdot), s(\cdot)))_{y \in [K]} \mid g : \mathbb{R}^d \to \mathbb{R}, s : \mathbb{R}^d \to (0, \infty)\}
$$
$$
\subseteq \{(P_{\text{acl}}(y; \{\rho[\boldsymbol{b}] - g(\cdot) \cdot \mathbf{1}\}/s(\cdot)))_{y \in [K]} \mid \boldsymbol{b} \in \mathbb{R}^{K-1}, g : \mathbb{R}^d \to \mathbb{R}, s : \mathbb{R}^d \to (0, \infty)\}. \tag{47}
$$

*Proof of Theorem 15.* For $\boldsymbol{b} = (b_k)_{k \in [K-1]}$ satisfying $b_k = \log(k)/s$ for $k = 1, \ldots, K - 1$, $v = \log(u)/s$, and $\boldsymbol{w} = (w_k)_{k \in [K-1]} = \boldsymbol{b} - v \cdot \mathbf{1}$, one has that

$$
P_{\text{acl}}(y; \boldsymbol{w}) = \frac{\prod_{l=1}^{y-1} e^{-w_l}}{\sum_{k=1}^{K} \prod_{l=1}^{k-1} e^{-w_l}} = \frac{e^{-\sum_{l=1}^{y-1}\{\log(l)-\log(u)\}/s}}{\sum_{k=1}^{K} e^{-\sum_{l=1}^{k-1}\{\log(l)-\log(u)\}/s}} = \frac{e^{\{(y-1)\log(u)-\log((y-1)!)\}/s}}{\sum_{k=1}^{K} e^{\{(k-1)\log(u)-\log((k-1)!)\}/s}}, \tag{48}
$$

which is equal to $P_{\text{poi}}(y; u, s)$ in (8). This proves (20). Also, since $(\log(k)/s)_{k \in [K-1]}$ is ordered, (20) proves the latter results. □

## C   Supplement of Experimental Results

We here show details of the experimental results. Figures 7, 8, 9, and 10 respectively show errorbar-plots of mean and STD of the test NLLs, MZEs, MAEs, MSEs for all methods and datasets.

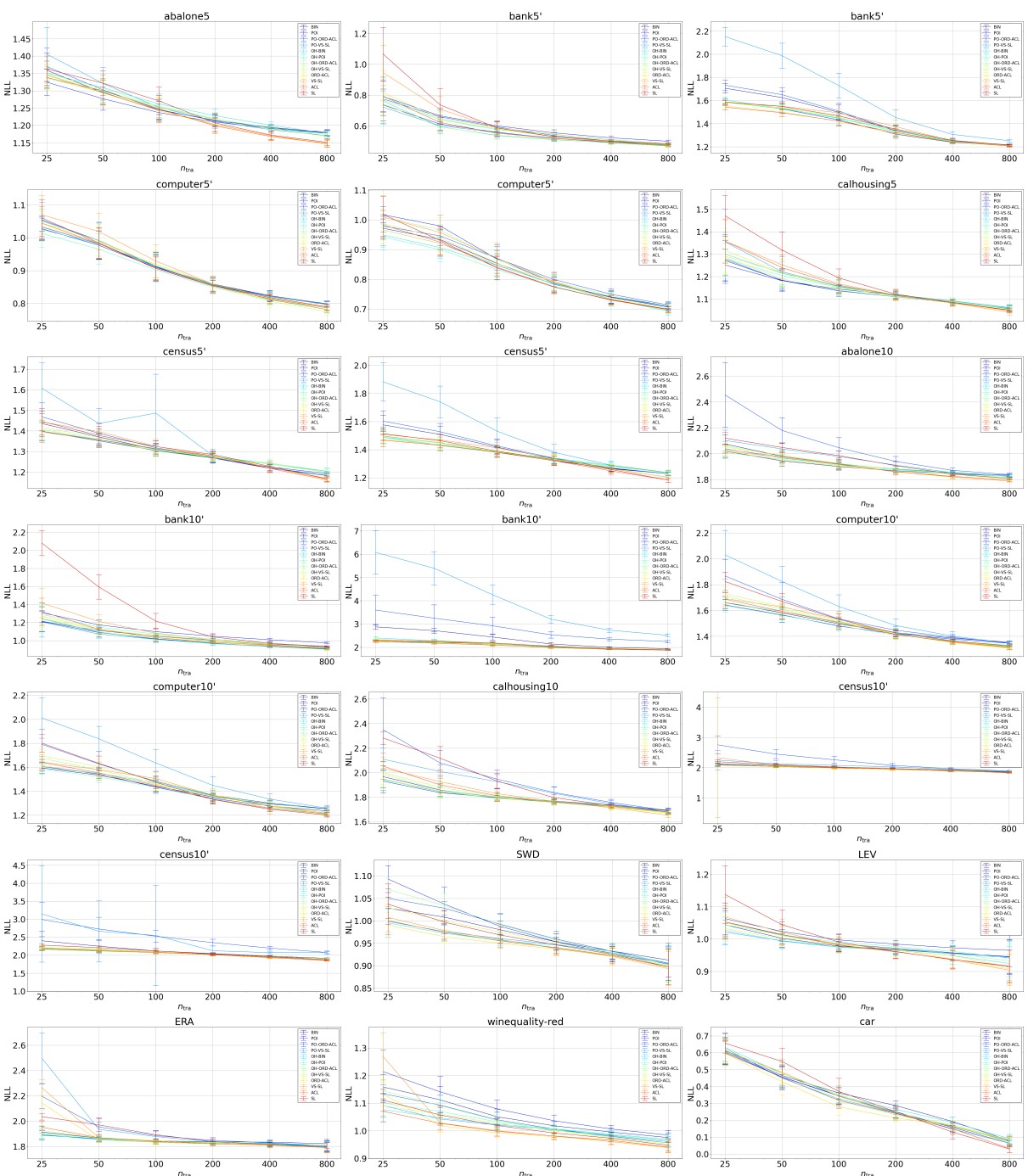

Figure 7: Errorbar-plots of mean and STD of the test NLLs of all methods for each dataset.

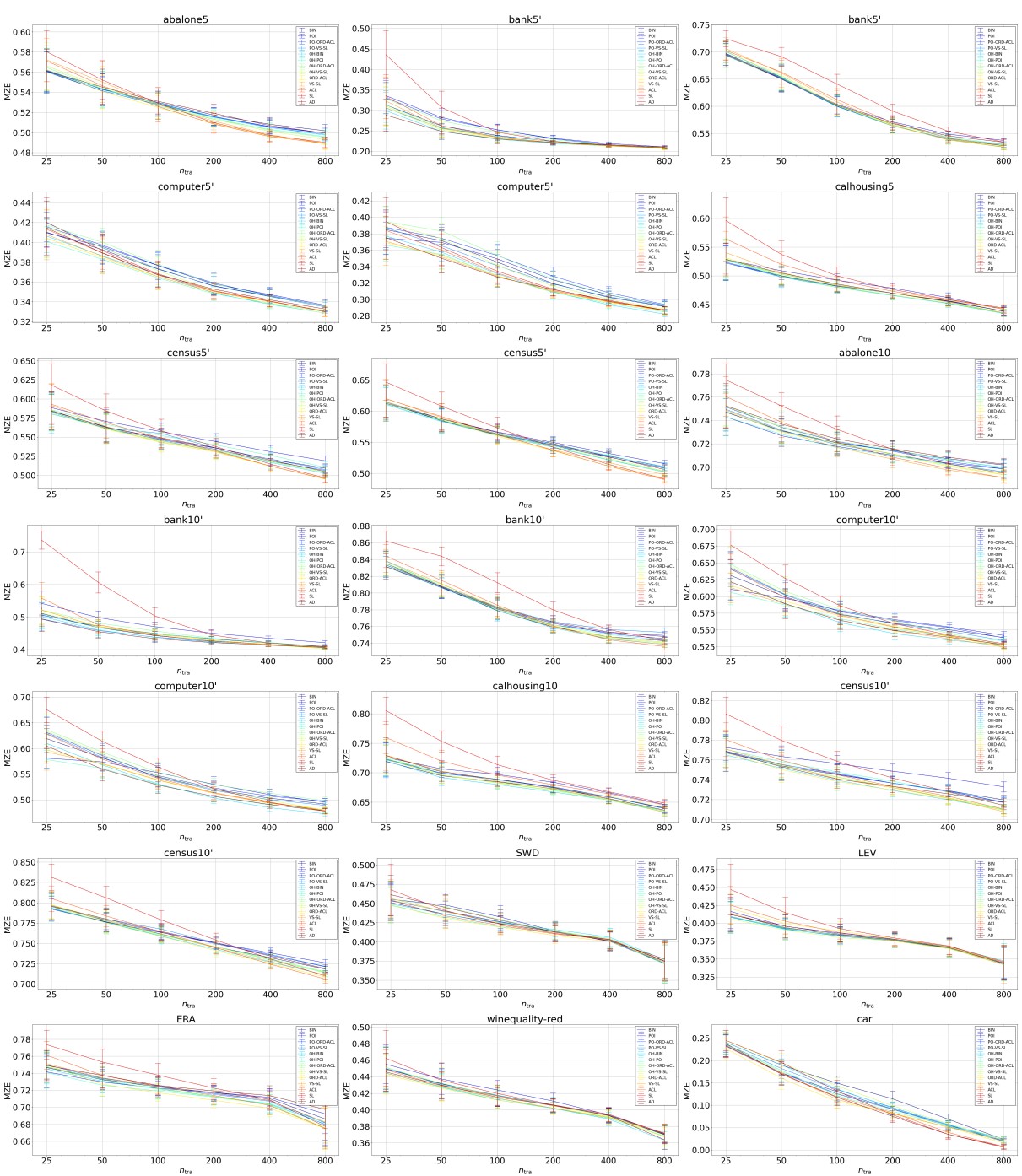

Figure 8: Errorbar-plots of mean and STD of the test MZEs of all methods for each dataset.

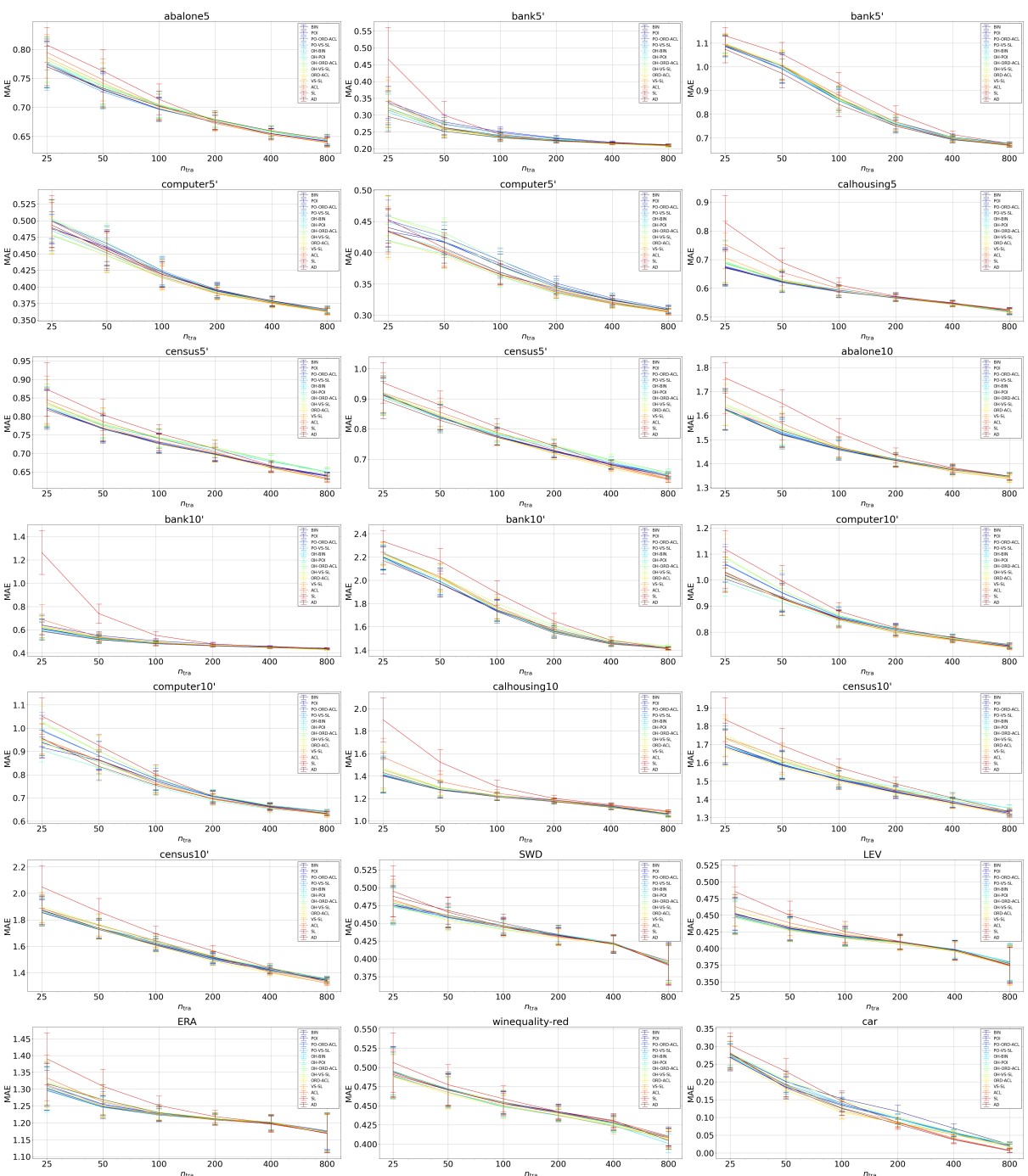

Figure 9: Errorbar-plots of mean and STD of the test MAEs of all methods for each dataset.

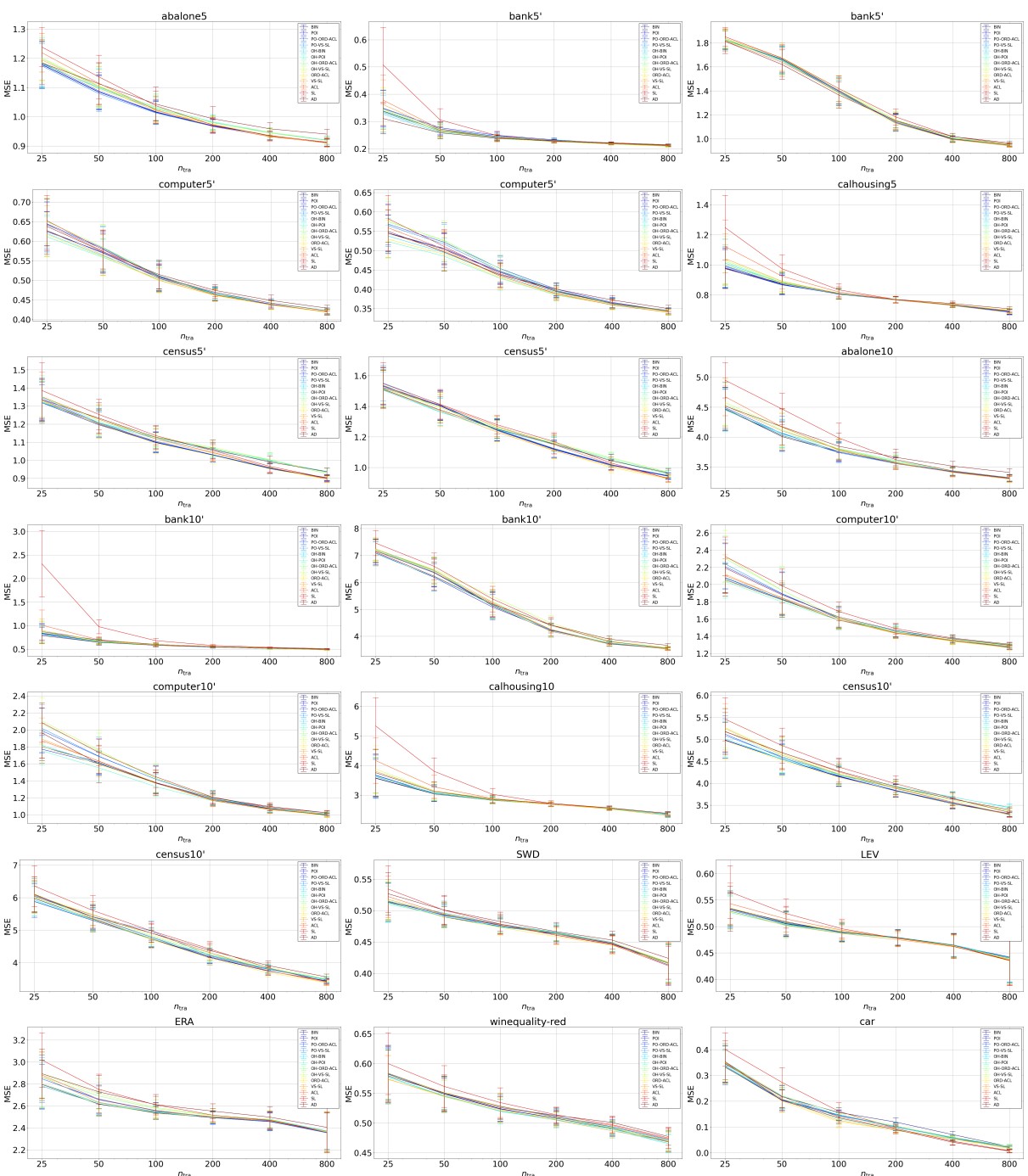

Figure 10: Errorbar-plots of mean and STD of the test MSEs of all methods for each dataset.

