# OpenReview forum: "Unimodal Likelihood Models for Ordinal Data"
_TMLR — Accepted by TMLR_

### Review · Reviewer_CPCb · 2022-07-24

**Summary Of Contributions:**

This paper introduces different models fitted to ordinal regression, when one wants the predicted output conditional probability to be unimodal, an assumption that may seem responsable in ordinal classification contexts (although one may easily image input samples for which there is a polarised response, such as a movie that most people either love or hate).

Properties of the new models are discussed, as well as their associated flexibility. Finally, some experiments are performed (using neural networks as ways to estimate the mapping from input space to predictive probabilities) to show the behaviour of the different models, under different training regimes.

**Broader Impact Concerns:**

No Broader Impact statement needed.

**Requested Changes:**

I will denote my most pressing requests by a plus **(+)** in front of the label. Note that addressing the most pressing ones would most probably make the paper well over 12 pages.

* Section 2.1., end of second paragraph: I do not understand the point of mentioning that ordinal data cannot be more rigorously defined? Ordinal data are what they are, and their rigorous definition is to be ordered categories. What more would be needed?

* Definition 3 and definition 4: here, authors provide some characterisation of the distributions. However, while some qualitative comments are indeed given when studying each models, such characterisation and their impact in possible experiments is not discussed further. Similarly, in the definition, authors speak of heterogeneity or homogeneity of these features across the input space, but do not provide any (statistical) test to see if this is the case in data sets, nor do they discuss such aspects in the experiments where real data sets are used.

* Definition 5 is a very strange way to state that the DoF of a parametric model correspond to the number of parameters it depends on? Why invoking manifold here and the function $g$, when we are speaking about a probability vector?

* **(+)** Theorem 1 and everywhere in the paper where formal results are stated: while I acknowledge that the result may not be difficult to prove, A formal statement in a paper (and even more in a journal paper) should definitely be proven in order to check the validity of the paper. If I am correct, papers in TMLR are supposed to be self-contained papers, which is not the case if proves of statements are missing. If space does not allow it, then authors can still go to a venue where space is unconstrained.

* **(+)** Theorem 2 really needs a proof, even more so than theorem 1. Also, it would be desirable  to explicitly state the condition mentioned just before Theorem 2, in the style "If The ACL model satisfy THIS condition, then it is a unimodal distribution". Right now it is totally similar to theorem 1, and I cannot really identify the mentioned condition, especially as this condition seems to have to do with the $\'{g}$ function defined later, that is absent from the theorem?

* **(+)** P7, top: it is stated that the ORD-ACL generalise both the POI and BIN model. This would need to be proven, or at least to mention for which choices of parameters in ACL we do retrieve these two models (leaving the exercice to the reader to check it, even if a proof wold be better). I get that theorem 3 somehow states it for ORD-ACL and POI, but the theorem is again not proved.

* **(+)** Why is the presentation for the VS-SL model not following the same format than for the previous models? I gather that the added flexibility comes from the additional V transformation, nevertheless I would have expected unimodality, decrease rate etc. to also be expressed as theorems.

* **(+)** In the experiment, I will not really question the considered model/neural network (as this is beyond my scope of expertise), however I am wondering what is the use of using the probabilities to minimise a given loss, and then to assess the models according to another metric? For example, why not using the MAE only when considering $\ell_{abs}$, for instance?

* Finally, one could wonder why proposing different models with extra degrees of flexibility (such as the $\phi$ or $\tau$ functions, if it is to choose fixed ones in the experiments (for instance, the added flexibility of the VS-SL model does not seem to be explored here?)

* **(+)** Last but not least, the paper english could largely be improved, as several places contains forms of sentences that are not proper english. For example (this is not an exhaustive list):
- Models are expected promising (abstract)
- Our took OR experiments (abstract)
- Symplex --> simplex (Definition 1)
- The readers should recognise our treatment of them with reference (P3)
- To every our treated likelihood-models (P4)

**Strengths And Weaknesses:**

+: a set of new models (maybe too many?) from which to choose in order to perform ordinal regression
+: some interesting discussions about these model properties
+: a comprehensive experimental studies, considering many models and settings

-: no proof of provided theoretical results
-: difficulty to really know when to apply which model, or one needs so many alternatives and what problem they solve
-: unclear whether such a large experimental setting is needed (why using a given metric to predict and another to measure the prediction quality?)

---

> ### Author Response · Authors · 2022-07-26
> **1st Reply to Review by Reviewer CPCb**
>
> We greatly appreciate the quick and constructive review.
> We here reply pointed Weaknesses and Requested Changes.
> But, we wait until all 3 reviews have been submitted before submitting a revised PDF, as the recommendation by TMLR.
>
> [-:no proof ~] & [(+)Theorem 1 ~]
> We will add proofs for theorems to the appendix.
>
> [-:difficulty to ~]
> This issue is not limited to our study, but is extremely common:
> We do not know whether we should use logistic or probit regression, or SVM (hinge-loss) before trying.
> However, when a model A performs poorly and a more representable model B performs well, it would be promising to consider an "intermediately representable between A and B"-model C and a "further more representable"-model D, in general.
> Our discussion helps such model selection.
>
> [Section 2.1. ~] & [-:unclear whether ~] & [(+)In the experiment ~]
> The term "ordered" is not mathematical nor quantitative.
> Some readers may think this is non-rigorous.
> We have to mention that a more rigorous definition does not exist even in previous studies.
> However, we define ordinal data as the data discussed in previous OR studies, which would be expected to be ordered and categorical.
> This was a last resort.
>
> Due to the absence of mathematical definition of the ordinal data, we consider that we have to take large (using various datasets) experiments.
>
> Also, NLL would be standard for learning a statistical model, and OR tasks are usually defined as task risk minimization.
> We believe that the use of NLL, MZE, MAE, and MSE is correct and theoretically non-strange.
>
> There is no one-to-one relationship between task risk and optimized NLL, in general.
> In this paper, we consider constrained likelihood models.
> It is possible that such constraints may be a special cause of the phenomenon that task risks for models A and B with similar NLL differ greatly.
> Also, it is possible that a model A has good compatibility with Task-Z, but has bad compatibility with Task-S.
> We conducted large-scale experiments to experimentally deny such suspicions.
>
> [(+)Theorem 2 ~]
> We have written "Assume that $u_0(:=-\infty)\le\cdots\le u_{m-1}\le 0\le u_m\le\cdots\le u_K(:=+\infty)$ for some $m\in[K]$." in Theorem 2.
> The ordering condition $u_1\le\cdots\le u_{K-1}$ is related to the ordered model $\acute{{\bf g}}$.
> We will emphasize this relation and rephrase the statement to "If ${\bf u}\in R^{K-1}$ satisfies $u_0(:=-\infty)\le\cdots\le u_{m-1}\le 0\le u_m\le\cdots\le u_K(:=+\infty)$ for some $m\in[K]$, it holds that ~".
>
> [(+)P7, top ~]
> The ORD-ACL model can represent all data with a unimodal CPD that has DRs' unimodality constraint.
> This fact implies that the ORD-ACL model generalizes both the POI and BIN models.
> As a non-trivial result, Theorem 3 shows that the "PO"-ORD-ACL model generalizes the POI model.
>
> Regarding ORD-ACL and BIN models:
> see (4) and (9);
> $P_{bin}(y;u,s)=P_{acl}(y;{\bf v})$ with ${\bf v}=(v_k)\in R^{K-1}$ satisfying $v_k=\{\log(P_{bin}(k;u))-\log(P_{bin}(k+1;u))\}/s$;
> $v_1\le\cdots\le v_{K-1}$;
> These show the generalization.
>
> However, note that the statement "the PO-ORD-ACL model generalizes the BIN model" is incorrect in general.
> We will remark this fact, at the end of Sec.5.1.
>
> [(+)Why is the presentation ~]
> The SL model can represent all data, while the VS-SL model can represent any data having a unimodal CPD.
> Namely, the VS-SL model reduces the representation ability from the SL model.
>
> The SL model $P_{sl}(y;{\bf g}({\bf x}))$ can be formulated with ${\bf g}:R^d\to R^{K-1}$ equivalently, but the VS-SL model cannot:
> The (K-1)-DF requires at least (K-1) elements, and it requires an additional 1 element to decide where is the mode.
> Thus we formulated the SL link function $P_{sl}(y;{\bf u})$ with ${\bf u}\in R^K$.
>
> Also, since the VS-SL model can represent all data with a unimodal CPD, additional DRs' constraint is none.
>
> [Definition 3 ~]
> For notions related to the scale and skewness, we introduced numerical measures in Definitions 3 and 4 just for the sake of clarity of the qualitative discussion.
> However, we did not introduce metrical meaning into these measures (and it is quite difficult):
> for example, a gap between scales 0.1 and 0.3 and a gap between scales 0.7 and 0.9 do not have the same meaning.
> Therefore, we can not evaluate the heteroscedasticity or mode-wise skewness without misleading.
> We will add this reason to the footnote.
>
> [Definition 5 ~]
> We defined the DF for functions, not as something that is often used for point estimates in statistics.
> To avoid the misleading, we will rename it functional DF (FDF).
> "manifold" is mathematically natural for DF:
> see "Statistics and mathematics" in https://en.wikipedia.org/wiki/Degrees_of_freedom
> We add easier explanation of FDF after Def.5.
>
> [Finally, one could ~]
> Different selections of reasonable functions $\rho$ and $\tau$ do not change the representation ability of the VS-SL models qualitatively, but change the representation ability of the PO-VS-SL and HO-VS-SL models (a little).

---

### Review · Reviewer_ofxX · 2022-08-02

**Summary Of Contributions:**

This is a paper about the classification by probabilistic methods with ordinal target variables (i.e., there is an ordering relation in the set of possible values and we expect the corresponding conditional distribution to reflect this order by being unimodal. The authors derive a (mostly qualitative) characterisation of such notion of unimodality and show by extensive experiments that existing approaches on likelihood models based on binomial and Poisson distributions are less effective than the ones presented here.

**Broader Impact Concerns:**

No specific issues.

**Requested Changes:**

I would definitely add to the paper a related work section where all the existing approaches in the field are discussed.
In the experimental section I would better use a summary to better outline the main conclusions.
A toy example to be considered over the different sections of the paper would help (while the figures are often a bit hard to follow).

**Strengths And Weaknesses:**

Strengths
- The problem of taking advantage of the ordinal nature of the target variable seems to be very interesting.
- The experimental analysis sounds quite strong to me. Many alternative methods are considered and the results clearly advocate the advantages of the new methods.

Weaknesses
- I am not an expert of this field, but I have the impression that not all the existing literature is discussed.
- The (qualitative and empirical) justification of the advantages of the proposed approaches might appear a bit weak
- The discussion of the results is very hard to follow

---

> ### Author Response · Authors · 2022-08-02
> **1st Reply to Review by Reviewer ofxX**
>
> We greatly appreciate the quick and constructive review.
> We here reply pointed Weaknesses and Requested Changes.
> But, we wait until all 3 reviews have been submitted before submitting a revised PDF, as the recommendation by TMLR.
>
> [I am not an expert ~] & [I would definitely ~]
> Ordinal regression studies discussing on the unimodal likelihood models are really few.
> (The fact that they are few does not mean that they are less important. There is also an ICML paper.)
> We could find only groups of "Joaquim F Pinto da Costa, Hugo Alonso, and Jaime S Cardoso" and "Christopher Beckham and Christopher Pal".
> Thus, we discussed in Section 3 all existing unimodal models that we were aware of.
> The group of "Maria Iannario and Domenico Piccolo" reviewed in Section 5.3 discusses a clear focus on mixed models rather than unimodality.
> We believe that this paper discusses most of major researches on unimodal likelihood models.
>
> [The discussion of ~] & [In the experimental ~]
> We agree to use a summary to better outline the main conclusions in the experimental section.
> We will add it following your advice.
>
> [The (qualitative and ~]
> Compare our experiments with "Joaquim F Pinto da Costa, Hugo Alonso, and Jaime S Cardoso. The unimodal model for the classification of ordinal data. Neural Networks, 21(1):78–91, 2008." and "Christopher Beckham and Christopher Pal. Unimodal probability distributions for deep ordinal classification. In International Conference on Machine Learning, pp. 411–419, 2017.".
> We believe that our experiments will be larger (regarding variety of datasets, tasks, and training sample size) and more convincing, and that we provided much qualitative and empirical justification.
> Experimental summary in revised version will improve the understanding.
> Note that theoretical justification of unimodal model has not exist in the literature yet.
>
> [A toy example to ~]
> Figures 2-5 are popular representation in the ordinal regression research.
> For example, refer to "Fig. 1" of "John A Anderson. Regression and ordered categorical variables. Journal of the Royal Statistical Society: Series B (Methodological), 46(1):1–22, 1984.".
> We expect that these figures will play a role like toy-examples that you have pointed out.

---

### Review · Reviewer_qPrL · 2022-08-03

**Summary Of Contributions:**

The authors studied unimodal models in the context of ordinal regression. Through a set of experimental results they showed that their "more representable" unimodal models obtained a better generalization performance, compared to previous unimodal likelihood models.

**Broader Impact Concerns:**

No concerns on ethical implications of the work.

**Requested Changes:**

1. Motivations for introducing unimodality in OR tasks;
2. More systematic analysis on what "representability" means;
3. More rigorous results on why representability is important for obtaining good generalization performance for OR tasks;
4. Design experiments to verify your theoretical findings, if any.
5. Correcting several typos in the paper.

**Strengths And Weaknesses:**

The theoretical contribution is limited. The authors used quite some space showing existing definitions and results, but did not do a good enough job explaining the motivation / intuition behind using unimodal likelihood functions for ordinal regression tasks.

The experiments might possess some value, in the sense that it might point to some useful directions for future research. But the limited theoretical contributions make it disqualified for publication at TMLR.

---

> ### Author Response · Authors · 2022-08-04
> **1st Reply to Review by Reviewer qPrL**
>
> We greatly appreciate the quick and constructive review.
> We here reply pointed Weaknesses and Requested Changes.
>
> ---
> [but ~ tasks.] & [Motivations ~ tasks;] & [More ~ means;] & [More ~ tasks;]
>
> Most ordinal regression studies do not explain the natural ordinal relation more specifically, as we described in "Sec.2.1,Para.2".
> We introduced unimodality as a working hypothesis, believing that a more specific characterization was needed for the systematic discussion.
>
> Considering your indication, we add the following explanation to "Sec.1,Para.3":
> We explain in more detail the relationship between the representation ability and generalization performance of statistical models, using bias-variance decomposition (and tradeoff).
> Weakly representable models (e.g., previous unimodal likelihood models) enlarge bias-dependent term of the generalization error, if they cannot represent the underlying data distribution.
> Unnecessarily flexible models (e.g., SL a.k.a. multinomial logistic regression model) can enlarge variance-dependent term of the generalization error, especially when the training sample is small.
>
> Although we did not widely appeal it, existing ordinal regression studies, including (da Costa et al.,2008; Beckham & Pal,2017), have never directly verified that many ordinal data satisfy almost-unimodality (fact*), and our experiments in Sec.6.2 provide important novel knowledge.
>
> The bias-variance tradeoff and the fact* motivate our consideration of strongly representable unimodal likelihood models that adequately represent many ordinal data.
>
> ---
> [The ~ results,]
>
> At least the notions of "decay rates", "mode-wise/overall heteroscedasticity", and "mode-wise/overall skew" are those that this paper first introduced.
> Most existing ordinal regression studies have little discussed the representation ability of the models.
> (Beckham & Pal,2017), which extends (da Costa et al.,2008), also does not sufficiently discuss the differences in the representation abilities of their models.
> We believe that Def.5 and Qualitative notions that we introduced will have an important impact on ordinal regression research and great advance.
>
> ---
> [The ~ limited.] & [But ~ TMLR.] & [More ~ tasks;]
>
> Existing ordinal regression studies also lack meaningful (to compare methods) theoretical analysis except for that only for simple methods.
>
> It is very difficult to perform generalization analysis on parametrically constrained classification methods.
> Refer to, e.g., the end of the Section 4 of "Pedregosa, Fabian, Francis Bach, and Alexandre Gramfort. On the consistency of ordinal regression methods. JMLR 18.1 (2017): 1769-1803.".
> We consider that their analysis and ours are not very different in a technical level essentially (although their and our presentations/considerations are different).
> Note that Fabian Pedregosa is Managing Editors of TMLR, and the above paper is submitted to JMLR (we are not related to them!).
> Namely, we hope that reviewers will understand the difficulties inherent in ordinal regression studies.
>
> We also believe that we have done everything technically possible at this time, and that our results are novel and unique in that they allow us to compare methods (especially in ordinal regression research).
> Our qualitative discussions will play a big rule in comparing and developing OR methods.
>
> We also considered (though not in the paper) an analysis based on first-order asymptotics, assuming a linear model in classical statistics.
> For example, under the assumption $g(\bf{x})=\bf{\beta}^\top \bf{x}$, PO-ORD-ACL and PO-VS-SL models can be analyzed but are not comparable.
> Our discussion is to compare methods.
> Quantitative analysis that allows comparison is difficult (typical generalization analysis does not allow comparison).
>
> We suspect that our poor description has not conveyed the contributions of this study well.
> However, our revision will improve this issue.
> If the reviewer still requires further analysis after reading the revision, please write what exactly you want us to show.
> We will consider it.
>
> ---
> [Design ~ any.]
> Compare our experiments with those by (da Costa et al.,2008) and (2011; Beckham & Pal,2017).
> Ours will be larger (regarding variety of datasets, tasks, and training sample size) and more convincing, and that we provided much qualitative and empirical justification.
> We believe that ours are succeeded to verify the suggestion from theories:
> Likelihood models with weak (resp. strong) representation ability are more effective when the training sample size is small (resp. large), and unimodal models were better than ACL and SL models which are more representable but have no unimodality guarantee.
>
> We added Figure 6 (in Sec.6.2), and changed to wrote the outline of results first and the detailed considerations next (in Sec.6.3).
> This change will improve the readability of the paper, and we are confident that the reviewers will be satisfied.
>
> Or, to prove (verify) our theorems, we add proofs in App.A.

---

### Author Response · Authors · 2022-09-03
**Additional Claim**

Although most existing OR studies have not gone into "what is a natural ordinal relation", we expect that the unimodality hypothesis (whose validity is first confirmed in this paper), and especially the VS-SL model which can represent arbitrary unimodal data will play an important role in future OR studies (in a theoretical perspective).
We turned to want to emphasize this point additionally.

Of course, the ORD-ACL models are also promising in practical perspective.

---

### Decision · Action_Editors · 2022-09-14

**Recommendation:** Accept with minor revision

**Comment:**

This paper proposes several novel models for ordinary regression (OR), which interpolates existing methods in terms of representation ability.  The authors defined several measures of the model's property, with which flexibility of models are theoretically analyzed.  Experimental results with several datasets demonstrate the usefulness of proposed models.

Most reviewers acknowledged the contributions of the paper, and recommended acceptance of the revision.  I agree with them that the paper contains significant contributions, and the paper should be published.  However, the revision still seems unnecessarily hard to read.  Therefore, I suggest the following modifications before publication.

- I see Figure 6 very important and it can make readers easily understand the main contributions of the paper.  This figure explains which kind of models were missing before this paper, and readers will acknowledge that this paper fills many nodes of the tree structure of models with theoretical justification and empirical performance improvement.
To make full use of this figure, I suggest the following modifications:

1. This figure should already be referred to in Section 5, where many models are introduced and the relation is explained.  I strongly felt that this kind of figure is necessary when reading Section 5, and found it later in Section 6.  The authors could show this figure even in the introduction to motivate readers, by saying that there are many missing models in the tree, and the current paper fills them with new models, and will be later shown to outperform existing methods in some situations.
2. Visually distinguish between the proposed new models and the existing models (for example, bold-faced for the new models and the normal font for the existing models.  This should be mentioned in the caption).
3. If I understand correctly, all the 11 arrows indicate the superset/subset (SS) relationship (i.e., the children are special cases of the parents for each arrow).
Then, it would be much easier to understand the contributions of the paper if the authors provide the 11 theorems that state the SS relation of all 11 arrows, and prove all of them in the appendix.  For proving SS relation, you can simply shows what set of the parameter values corresponds to a child model.  Showing the theorem numbers next to each arrow in Fig.6 would be also helpful.  The SS relations might be trivial for the authors but not for readers, so explicit statements will improve the readability.

- In Sections 4 and 5, many claims are made as plain text, and it's not clear if they are mathematically rigorous claims or some intuitive, non-rigorous arguments.  For example, "ORD-ACL fully encompass the BIN and PO models", "VS-SL models can be viewed the most representable ones among unimodal likelihood models..."  If they are rigorous claims, it should be stated as theorems and proved.  To prove "VS-SL models can be viewed the most representable", you could, for example, define the region spanned by the most general unimodal models in the density function space, and show that it matches the image of the parameter domain of VS-SL onto the function space.   When you claim for example that "The PO-ORD-ACL model generalizes the BIN model" is incorrect in general, you should also prove it by, for example, showing that a particular instance of BIN cannot be expressed as PO-ORD-ACL.  Those things are not trivial for all readers.

- Definition 5 (the term "stronger representation ability" can be only used for the superset relation) is confusing because this term tends to be used more casually when comparing two models which are not in the SS relation.  Why don't you simply says model 2 is a special case of model 1?  You could also use the SS terminology.

- I don't understand Definitions 3, 4, and 6, which must be rewritten.  What do you mean by "zero irrelevant"? Why f suddenly appears?  For Definition 6, why don't you define FDF as the manifold dimension of the model?

- "The PO-ORD-ACL and PO-VS-SL models are novel. The significance of these models we expect lies in the theoretical guarantee of the unimodality, not in the improvement of probability prediction and classification performances"
If this would be true, the authors should explain why unimodal guarantee is useful even if it won't improve the performance.
I think the authors think that they could improve the performance, thanks to the unimodality on some data, which the authors didn't find.
If so, modify the sentence.

- I expect that many readers of TMLR are not expert of OR, and they (including myself) would wonder what happens if we simply train a regressor with the ordered labels treated as continuous values, e.g., {'strongly agree' = -2, 'agree' = -1, 'neutral' = 0, 'disagree' = 1, 'strongly disagree' = 2}, and discretize the output in the test phase.  Mentioning this point in the introduction would be helpful for non-experts.  Including this regressor as a weak baseline in one of the experiments would be also nice.

- As motivation, the authors argue that statisticians implicitly expect the unimodality, and showed that this expectation is right by analyzing existing datasets.  However, the unimodality can be easily violated by combining bins: even if the labels {'strongly agree', 'agree', 'weakly agree', 'neutral', 'weakly disagree', 'disagree', 'strongly disagree'} are unimodal, a different binning {'agree' = {'strongly agree', 'agree', 'weakly agree'}, 'neutral', 'disagree' = {'weakly disagree', 'disagree', 'strongly disagree'}}  can be not unimodal.  In this sense, one can't say most data is expected to be unimodal because most benchmark data is unimodal.  It should be cautioned that unbalanced grouping of the labels can violate the unimodality, and in such cases the proposed unimodal methods shouldn't be used.

- In Table 3, "ranking of the methods" are shown.  I wonder why don't you show for example average NLL, MAE,..., and apply for example the Wilcoxon signed rank test and highlight the best methods and those which are not significantly outperformed by the best method?  If showing ranking is standard in OR, the authors don't need to change but explain why showing ranking is better with citations.

- MUs are shown Table 1. Can you show MU of samples drawn from a model (for example SL model) without unimodality, which could help demonstrate that SL is not a good model to fit the datasets in the table.


Minor but important comments:
- The paper is understandable, but the choice of words is not very typical, which degrades readability.  I strongly recommend the authors to ask someone (or some company) for proof-reading.

1. In the last paragraph of Section 1, "We took experimental comparisons" -> "We perform experimental comparisons"
2. "As we show the experimental results and considerations in Section 6.3 and Appendix B, we confirmed that ..."-> "Our empirical results show that"
3. The last paragraph of Section 2 is hard to read.  I understand it very vaguely.
4. Footnote 3 in Page 5: "meaningless"->"irrelevant"
5. Captions in many tables and figures:  "It shows the ***".  The first sentence in captions can be the title of the tables/figures, so remove "It shows".
6. In the beginning of Section 6.3: What is "simultaneous comparison"?
7. In Section 6.3, having "detail of results" and "POI v.s ..." in parallel doesn't make sense.  Maybe "outline of results" and "detail of results" could be subsubsections?
8. Use "citet" when the reference is a part of the sentence.

Please consider all the comments above, and revise the paper accordingly.  If the authors disagree on some comments, please rebut.  Although many, most of my suggested modifications are about how to present the theoretical/empirical results that the authors already have.  Accordingly, my recommendation is accept with minor revision.

---

> ### Author Response · Authors · 2022-09-28
> **Thank you for the decision**
>
> Thank you for the decision.
>
> We wrote a deanonymized camera ready version.
> We consider that this version adopts most of the advices.
> Please see our treatments for each advice and "Changes Since Last Submission".